# Solving Inverse Problems with Flow-based Models via Model Predictive Control

George Webber [* 1]   Alexander Denker [† * 2]   Riccardo Barbano [3]   Andrew J Reader [1]

## Abstract

Flow-based generative models provide strong unconditional priors for inverse problems, but guiding their dynamics for conditional generation remains challenging. Recent work casts training-free conditional generation in flow models as an optimal control problem; however, solving the resulting trajectory optimisation is computationally and memory intensive, requiring differentiation through the flow dynamics or adjoint solves. We propose MPC-Flow, a model predictive control framework that formulates inverse problem solving with flow-based generative models as a sequence of control sub-problems, enabling practical optimal control-based guidance at inference time. We provide theoretical analysis linking MPC-Flow to the underlying optimal control objective and show how different algorithmic choices yield a spectrum of guidance algorithms, including regimes that avoid backpropagation through the generative model trajectory. We evaluate MPC-Flow on benchmark image restoration tasks, spanning linear and non-linear settings such as in-painting, deblurring, and super-resolution, and demonstrate strong performance and scalability to massive state-of-the-art architectures via training-free guidance of FLUX.2 (32B) in a quantised setting on consumer hardware.

## 1. Introduction

Continuous normalising flows (Chen et al., 2018), particularly those trained via flow matching (Lipman et al., 2023), have become an established and powerful framework for generating high-dimensional data (Esser et al., 2024; Wang et al., 2025b). These models define a continuous-time transformation through an ordinary differential equation (ODE) that maps samples from a simple base distribution to complex data distributions. Beyond unconditional generation, there is significant interest in leveraging the learned dynamics of these models to solve inverse problems. Here, the goal is to recover a signal $\boldsymbol{x} \in \mathbb{R}^d$ from noisy measurements

$$\boldsymbol{y} = \mathcal{A}(\boldsymbol{x}) + \boldsymbol{\epsilon}, \tag{1}$$

with $\mathcal{A}$ a (possibly non-linear) forward operator and $\boldsymbol{\epsilon}$ denoting additive Gaussian noise, i.e., $\boldsymbol{\epsilon} \sim \mathcal{N}(0, \sigma^2 I)$.

Recent work has explored training-free guidance methods that adapt pre-trained flow models to conditioning constraints at inference time (Kim et al., 2025; Pokle et al., 2024; Zhang et al., 2024), see also Appendix A for a discussion on related work. These approaches incorporate data fidelity terms or task-specific loss functions directly into the generative dynamics, steering the flow towards regions that satisfy the desired constraints without retraining the base model. Such approaches are often motivated by the Bayesian perspective on inverse problems (Stuart, 2010), where the prior $p_{\text{data}}(\boldsymbol{x})$, given by the pre-trained flow model, is combined with likelihood $p(\boldsymbol{y}|\boldsymbol{x})$. However, these heuristic guidance strategies have limitations (Chung et al., 2023; Patel et al., 2024; Barbano et al., 2025). They often lack theoretical guarantees, e.g., regarding consistency to the measurements, and can be computationally unstable.

As an alternative, Liu et al. (2023b); Wang et al. (2025a) recast conditional sampling in continuous-time flow models as a deterministic optimal control problem. While the optimal control perspective provides a principled objective and improved trade-offs between data fidelity and prior consistency, solving the resulting trajectory optimisation requires differentiating through the sampling dynamics, leading to prohibitive computational and memory costs, or costly evaluations of the adjoint equations.

We propose MPC-Flow, a model predictive control (MPC) framework (Garcia et al., 1989; Rawlings et al., 2020) that makes optimal control-based guidance practical at inference time. Rather than optimising the entire trajectory at once, as in classical optimal control, MPC-Flow decomposes the control into a sequence of short-horizon sub-problems. At each time step, a local control problem is solved, the re-

*Equal contribution †Work done while at University College London. [1]School of Biomedical Engineering and Imaging Sciences, King's College London [2]Helmholtz Imaging, Deutsches Elektronen-Synchrotron DESY, Germany [3]Department of Computer Science, University College London. Correspondence to: Alexander Denker <alexander.denker@desy.de>.

*Proceedings of the 43rd International Conference on Machine Learning*, Seoul, South Korea. PMLR 306, 2026. Copyright 2026 by the author(s).

sulting control is applied, and this process is repeated in a receding-horizon fashion. Our proposed framework allows for a control strategy that significantly reduces memory requirements by decoupling control optimisation from the full flow trajectory, while improving robustness through iterative re-planning that corrects errors accumulated at earlier time steps. Moreover, different choices of horizon length, control parameterisation, and terminal cost function give rise to a range of guidance algorithms within this unified framework, including an efficient and principled variant that avoids backpropagation through the generative trajectory.

We validate the proposed framework empirically across a range of controlled and large-scale settings. We first analyse MPC-Flow in controlled settings, including a two-dimensional toy example and a computed tomography task on the OrganCMNIST dataset, enabling detailed ablations and direct comparison with globally optimal control solutions. We then evaluate MPC-Flow on a range of linear and non-linear image restoration inverse problems on the CelebA dataset, and finally demonstrate scalability to modern large-scale rectified flow models.

To the best of our knowledge, MPC-Flow is the first conditional generation method explicitly rooted in optimal control that scales to such settings, enabling training-free guidance under tight memory constraints and outperforming existing scalable approaches, including guidance of FLUX.2 (32B) (Labs, 2025) in a quantised setting on consumer hardware.

## 2. Background

### 2.1. Flow-based Models

Flow Matching (FM) (Lipman et al., 2023) is a framework for training continuous normalising flows using a simulation-free objective. The goal is to learn a time-dependent vector field $v_\theta : \mathbb{R}^d \times [0,1] \to \mathbb{R}^d$ that induces a probability path $p_t$ transforming a simple base distribution $p_0$ (e.g., a standard Gaussian) into a data distribution $p_{\text{data}} := p_1$. Sample generation is performed by solving the ODE

$$\frac{d\boldsymbol{x}(t)}{dt} = v_\theta(\boldsymbol{x}(t), t), \qquad \boldsymbol{x}(0) \sim p_0. \quad (2)$$

Training proceeds by regressing $v_\theta$ toward a ground-truth velocity field $u_t$ that generates the desired probability path $p_t$, by minimising the flow matching objective

$$\mathcal{L}_{\text{FM}}(\theta) = \mathbb{E}_{t, \boldsymbol{x} \sim p_t} \big\| v_\theta(\boldsymbol{x}, t) - u_t(\boldsymbol{x}) \big\|^2.$$

However, the marginal velocity field $u_t$ is generally intractable to compute, as it depends on the unknown intermediate density $p_t$ and its score for complex high-dimensional data distributions. To address this intractability, FM adopts a conditional formulation. Conditional Flow Matching trains

the model to match a *conditional* vector field $u_t(\boldsymbol{x} \mid \boldsymbol{x}_0, \boldsymbol{x}_1)$, with $(\boldsymbol{x}_0, \boldsymbol{x}_1) \sim p_0 p_1$, by minimising

$$\mathcal{L}_{\text{CFM}}(\theta) = \mathbb{E}_{t, \boldsymbol{x}_0, \boldsymbol{x}_1, \boldsymbol{x}} \big\| v_\theta(\boldsymbol{x}, t) - u_t(\boldsymbol{x}|\boldsymbol{x}_0, \boldsymbol{x}_1) \big\|^2, \quad (3)$$

Crucially, it has been shown that $\nabla_\theta \mathcal{L}_{\text{CFM}}(\theta) = \nabla_\theta \mathcal{L}_{\text{FM}}(\theta)$ (Lipman et al., 2023; Tong et al., 2024). In practice, the conditional probability path is often defined via an affine interpolation between a source sample $\boldsymbol{x}_0 \sim p_0$ and a target data point $\boldsymbol{x}_1 \sim p_1$. A common choice is the linear interpolation used in rectified flows (Liu, 2022), $\boldsymbol{x}_t = (1-t)\boldsymbol{x}_0 + t\boldsymbol{x}_1$. This induces a simple conditional flow with velocity $u_t(\boldsymbol{x}|\boldsymbol{x}_0, \boldsymbol{x}_1) = \boldsymbol{x}_1 - \boldsymbol{x}_0$. Substituting this into (3) yields the simplified training objective

$$\mathcal{L}(\theta) = \mathbb{E}_{t, \boldsymbol{x}_0, \boldsymbol{x}_1} \big\| v_\theta(\boldsymbol{x}_t, t) - (\boldsymbol{x}_1 - \boldsymbol{x}_0) \big\|^2. \quad (4)$$

At inference time, the learned vector field $v_\theta$ is integrated according to (2) to generate samples from the model.

### 2.2. Optimal Control in Flow Models

Liu et al. (2023b); Wang et al. (2025a) frame conditional generation with pre-trained flow models as a deterministic optimal control problem, formalised as

$$\min_{\boldsymbol{u}} \left\{ \mathcal{J}(\boldsymbol{u}) = \int_0^1 \|\boldsymbol{u}(t)\|_2^2 \, dt + \lambda \, \Phi(\boldsymbol{x}(1)) \right\}$$
$$\text{s.t.} \quad \frac{d\boldsymbol{x}(t)}{dt} = v_\theta(\boldsymbol{x}(t), t) + \boldsymbol{u}(t), \quad t \in [0, 1]. \quad (5)$$

The trajectories are initialised at $\boldsymbol{x}(0) = \boldsymbol{x}_0$, sampled from the base distribution $p_0$. Here, $v_\theta(\boldsymbol{x}, t)$ denotes the pre-trained flow model, while the integral term regularises the control energy, encouraging trajectories to remain close to the nominal flow dynamics. The terminal loss $\Phi(\boldsymbol{x})$ encodes the conditioning objective. This optimisation problem seeks a control $\boldsymbol{u} : [0, 1] \to \mathbb{R}^d$ that steers the trajectory to minimise this terminal loss while maintaining minimal deviation from the pre-trained base model.

Under some standard conditions on the flow field $v_\theta$ and the terminal cost $\Phi$, we can prove the existence of an optimal control. The proof follows standard arguments and is provided in Appendix B.1.

*Remark* 2.1. For inverse problems, the terminal loss can be chosen as

$$\Phi(\boldsymbol{x}) = \frac{1}{2\sigma^2} \|\mathcal{A}(\boldsymbol{x}) - \boldsymbol{y}\|_2^2,$$

which corresponds to assuming additive Gaussian noise with variance $\sigma^2$ in the likelihood model; see (1). Under this choice, the resulting optimal control formulation admits an interpretation within the framework of variational regularisation (Engl et al., 1996). In particular, the induced regulariser is given by a distance induced by the flow model, measuring how costly it is to reach $\boldsymbol{x}$ from the initial point $\boldsymbol{x}_0$. We refer to Appendix H for a detailed discussion.

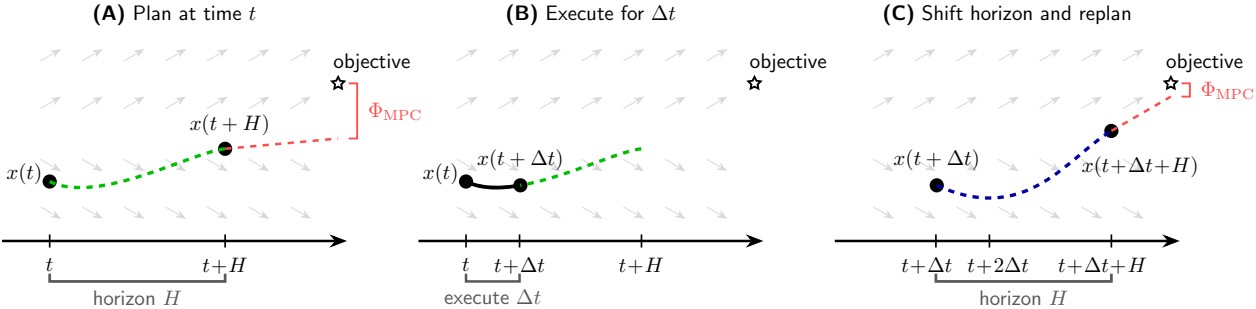

*Figure 1.* MPC-Flow strategy for guiding flow-based generative models toward a target objective. Starting from the current state, MPC-Flow plans a sequence of velocity adjustments that steer the flow toward the objective while keeping the intervention small (A). Only the initial part of the plan is applied (B), after which we re-plan from the new state and repeat the process (C).

In practice, the continuous-time optimal control problem in (5) is discretised. Both approaches proposed by Liu et al. (2023b) and Wang et al. (2025a), namely FlowGrad and OC-Flow, employ an explicit Euler discretisation with $N$ uniform time steps of size $\Delta t = 1/N$, leading to the following discretised problem:

$$\min_{\boldsymbol{u}_0,\ldots,\boldsymbol{u}_{N-1}} \frac{1}{N} \sum_{k=0}^{N-1} \|\boldsymbol{u}_i\|_2^2 + \lambda \Phi(\boldsymbol{x}_N)$$

$$\text{s.t.} \quad \boldsymbol{x}_{t_{k+1}} = \boldsymbol{x}_{t_k} + \frac{1}{N}(v_\theta(\boldsymbol{x}_{t_k}, t_k) + \boldsymbol{u}_k),$$

$$k = 1, \ldots, N-1, \quad t_k = k/N. \tag{6}$$

This discrete problem can, in principle, be solved using gradient descent, where the gradients for the controls $\boldsymbol{U} = [\boldsymbol{u}_0, \ldots, \boldsymbol{u}_{N-1}]$ can be computed using automatic differentiation by backpropagating through the ODE solver, or using the adjoint method (Chen et al., 2018), which requires simulation of two ODEs (the forward and the adjoint) for every gradient computation, increasing the computational cost (i.e., runtime). Automatic differentiation incurs a high memory usage that scales linearly with the number of time steps, since computing the gradient with respect to $\boldsymbol{u}_k$ requires $N - k$ nested evaluations of the flow model $v_\theta$, making it impractical for large-scale models. Check-pointing (Chen et al., 2016) is often employed to reduce memory usage, at the expense of increased computation time.

## 3. MPC-Flow

In this work, we propose MPC-Flow, a framework that leverages MPC (Garcia et al., 1989; Rawlings et al., 2020) for conditional generation with pre-trained flow models and addresses the computational bottlenecks identified in the previous paragraph.

MPC is an optimal control framework that has become standard in robotics and process control (Qin & Badgwell, 2003). At each time step, MPC computes a control by solving a finite-horizon optimal control problem initialised at the current state. Specifically, at time $t$, MPC considers a planning horizon $[t, t + H]$ and computes an optimal control $\boldsymbol{u}(\tau)$ over this interval. Rather than executing the entire trajectory, only the first part, corresponding to a short interval $\Delta t < H$, is applied to the system. The state is then updated, the horizon is shifted forward, and the optimisation is repeated. In this work, the MPC problem solved at time $t$ takes the form

$$\min_{\boldsymbol{u}} \int_t^{t+H} \|\boldsymbol{u}(\tau)\|^2 d\tau + \lambda \Phi_{\text{MPC}}(\boldsymbol{x}(t+H), t+H)$$

$$\text{s.t.} \quad \frac{d\boldsymbol{x}(\tau)}{d\tau} = v_\theta(\boldsymbol{x}(\tau), \tau) + \boldsymbol{u}(\tau), \quad \boldsymbol{x}(t) = \hat{\boldsymbol{x}}_t,$$

where $\hat{\boldsymbol{x}}_t$ is the current state at time $t$. MPC is defined by a choice of the horizon $H$, the number of discretisation steps $K$ used to partition the horizon, and an intermediate cost function $\Phi_{\text{MPC}} : \mathbb{R}^d \times [0, 1] \to \mathbb{R}_{\geq 0}$. The overall approach is illustrated in Figure 1.

In particular, we consider two distinct regimes: (i) receding-horizon control with $H = 1 - t$; and (ii) $\Delta t$-horizon control with $H = \Delta t$. These choices correspond to two extreme cases. For $H = 1 - t$, the control is optimised over the full remaining trajectory, whereas for $H = \Delta t$ the optimisation is restricted to the next intermediate time step. In the following paragraphs, we analyse these two design choices and summarise their key differences in Table 1.

### 3.1. Receding-Horizon (RHC)

We first analyse the time horizon $H = 1 - t$, namely receding-horizon control (MPC-RHC). MPC-RHC re-plans the entire remaining trajectory at every step,

$$\boldsymbol{u}^* \in \arg\min_{\boldsymbol{u}} \int_t^1 \|\boldsymbol{u}(\tau)\|^2 d\tau + \lambda \Phi(\boldsymbol{x}(1))$$

$$\text{s.t.} \quad \frac{d\boldsymbol{x}(\tau)}{d\tau} = v_\theta(\boldsymbol{x}(\tau), \tau) + \boldsymbol{u}(\tau). \tag{7}$$

The optimisation is carried out over $\tau \in [t, 1]$, with the state initialised as $\boldsymbol{x}(t) = \boldsymbol{x}_t$. The control is optimised over the remaining time horizon $[t, 1]$, but is only applied over a

---

**Algorithm 1** Receding-Horizon Control (RHC)

---

**input** Pretrained flow model $v_\theta(\boldsymbol{x}, t)$, terminal cost $\Phi(\boldsymbol{x})$, initial state $\boldsymbol{x}_0$, number of coarse steps $K$, step size $\Delta t$

1: $t' \leftarrow 0, \quad \boldsymbol{x} \leftarrow \boldsymbol{x}_0$
2: **while** $t' < 1$ **do**
3:     Set $t_k \leftarrow t' + k(1-t')/K$ # discretise $[t', 1]$
4:     Initialise control sequence $\boldsymbol{U} = [\boldsymbol{u}_0, \ldots, \boldsymbol{u}_{K-1}]$
5:     **Solve the finite-horizon optimal control:**

$$\boldsymbol{U}^* = \arg\min_{\boldsymbol{U}} \left[ \sum_{k=0}^{K-1} \frac{1-t'}{K} \|\boldsymbol{u}_k\|^2 + \lambda \Phi(\boldsymbol{x}_{t_K}) \right]$$

        subject to $\boldsymbol{x}_{t_{k+1}} = \boldsymbol{x}_{t_k} + \frac{1-t'}{K}\left(v_\theta(\boldsymbol{x}_{t_k}, t_k) + \boldsymbol{u}_k\right)$

6:     Apply only $\boldsymbol{u}_0^*$ on $[t', t' + \Delta t]$
7:     $\boldsymbol{x} \leftarrow \boldsymbol{x} + \Delta t\left(v_\theta(\boldsymbol{x}, t') + \boldsymbol{u}_0^*\right)$
8:     $t' \leftarrow t' + \Delta t$
9: **end while**

---

**Algorithm 2** $\Delta t$-Horizon Control

---

**input** Pretrained flow model $v_\theta(\boldsymbol{x}, t)$, MPC cost $\Phi_{\mathrm{MPC}}(\boldsymbol{x})$, initial state $\boldsymbol{x}_0$, step size $\Delta t$

1: $t' \leftarrow 0, \quad \boldsymbol{x} \leftarrow \boldsymbol{x}_0$
2: $\Phi_{\mathrm{MPC}}(\boldsymbol{x}, t) := \Phi(\boldsymbol{x} + (1-t)\, v_\theta(\boldsymbol{x}, t))$
3: **while** $t' < 1$ **do**
4:     Initialise control $\boldsymbol{u}$
5:     **Solve the one-step optimal control:**

$$\boldsymbol{u}^* = \arg\min_{\boldsymbol{u}} \left[ \|\boldsymbol{u}\|^2 + \lambda \Phi_{\mathrm{MPC}}(\boldsymbol{x}_{t'+\Delta t}, t' + \Delta t) \right]$$

        with $\boldsymbol{x}_{t'+\Delta t} = \boldsymbol{x}_{t'} + \Delta t\left(v_\theta(\boldsymbol{x}_{t'}, t') + \boldsymbol{u}\right)$

6:     Apply $\boldsymbol{u}^*$ on $[t', t' + \Delta t]$
7:     $\boldsymbol{x} \leftarrow \boldsymbol{x} + \Delta t\left(v_\theta(\boldsymbol{x}, t') + \boldsymbol{u}^*\right)$
8:     $t' \leftarrow t' + \Delta t$
9: **end while**

---

*Figure 2.* RHC optimises over the full horizon $[t, 1]$, discretised into $K$ steps; $\Delta t$-Horizon uses a 1-step lookahead horizon $[t, t + \Delta t]$. For MPC–RHC, the original terminal loss $\Phi$ is used, whereas MPC–$\Delta t$ employs the projected loss $\Phi_{\mathrm{MPC}}$.

short time horizon $[t, t + \Delta t]$ and the optimisation problem is re-solved from the updated state. As we solve the control problem up to the terminal time $t = 1$, the intermediate loss function is simply chosen to be $\Phi_{\mathrm{MPC}}(\boldsymbol{x}, 1) := \Phi(\boldsymbol{x})$.

To analyse the theoretical properties of this scheme, we make use of the value function $V(t, \boldsymbol{x})$, defined as the minimum cost-to-go from state $\boldsymbol{x}$ at time $t$,

$$V(t, \boldsymbol{x}) := \inf_{\boldsymbol{u}_{[t, 1]}} \left\{ \int_t^1 \|\boldsymbol{u}(\tau)\|^2 d\tau + \lambda \Phi(\boldsymbol{x}(1)) \right\},$$

where the infinum is taken with respect to the restricted interval $[t, 1]$. We can now formally state that the MPC-RHC recovers the global optimal control from (5).

**Theorem 3.1** (Optimality of Receding-Horizon Control). *Let $\boldsymbol{u}^*$ be the global optimal control for the problem (5). If at every time $t$, the sub-problem (7) with $H = 1 - t$ and $\Phi_{MPC}(\cdot, 1) = \Phi$ is solved to optimality, then the sequence of applied controls coincides with $\boldsymbol{u}^*$.*

The proof follows directly from Bellman's principle of optimality, see e.g. Liberzon (2011), and is provided in Appendix B.2.1 for reference. We validated this finding empirically with a 2D toy example, see Appendix C.

While Theorem 3.1 guarantees optimality, solving the MPC sub-problem over $[t, 1]$ remains, in principle, as expensive as solving the original problem (5). However, because we re-plan after every time step, MPC-RHC gains tractability through a coarse discretisation of the remaining horizon. At time $t$, we discretise $[t, 1]$ into $K$ steps with grid points $t_k = t + k(1-t)/K$ and approximate the controlled dynamics

*Table 1.* Comparison of MPC-Flow design choices.

| Method | Horizon $H$ | Discretisation | Backprop. through $v_\theta$ |
|---|---|---|---|
| **Receding-Horizon Control (RHC)** - Algorithm 1,3 | | | |
| $K > 1$ | $1 - t$ | $K$ steps | Yes ($\times K$) |
| $K = 1$ | $1 - t$ | 1 step | No |
| **$\Delta t$-Horizon Control** - Algorithm 2 | | | |
| MPC–$\Delta t$ | $\Delta t$ | 1 step | Yes ($\times 1$) |

using an explicit Euler scheme,

$$\boldsymbol{x}_{t_{k+1}} = \boldsymbol{x}_{t_k} + \frac{1-t}{K} \left[ v_\theta(\boldsymbol{x}_{t_k}, t_k) + \boldsymbol{u}(t_k) \right].$$

This recursion is applied for $k = 0, \ldots, K-1$, starting from $\boldsymbol{x}_{t_0} = \boldsymbol{x}_t$. As $t$ increases, the remaining horizon shortens and the effective step size decreases, naturally yielding finer temporal resolution near the terminal time while allowing coarser discretisation earlier in the trajectory. The method is summarised in Algorithm 1; however, for $K > 1$, this variant may be unsuitable for real-time deployment, as solving the resulting sub-problem still requires backpropagation through $K-1$ nested evaluations of the flow model $v_\theta$.

**Single-Step RHC (K=1).** A special case is to set $K = 1$, i.e., solving the ODE on the remaining interval $[t, 1]$ using a single time step. For this choice the problem reduces to

$$\min_{\boldsymbol{u} \in \mathbb{R}^d} \left\{ \mathcal{J}_{K=1}(\boldsymbol{u}) = (1 - t')\|\boldsymbol{u}\|^2 + \lambda \Phi(\boldsymbol{x}') \right\},$$

$$\text{with } \boldsymbol{x}' = \boldsymbol{x} + (1 - t')\left(v_\theta(\boldsymbol{x}, t') + \boldsymbol{u}\right),$$

This approximation effectively linearises the flow path. Crucially, the gradient $\nabla_{\boldsymbol{u}} \mathcal{J}_{K=1}(\boldsymbol{u})$ does not require backpropagating through the flow model $v_\theta$. This dramatically reduces GPU memory requirements and computational cost,

making the method particularly suitable for scaling to large state-of-the-art architectures. This special case of Algorithm 1 is given in the Appendix, see Algorithm 3.

## 3.2. $\Delta t$-Horizon Control

Alternatively, we consider the greedy limit where the planning horizon is minimal, i.e., $H = \Delta t$. In this regime, we only optimise the control for the next immediate time step,

$$
\begin{aligned}
\boldsymbol{u}^* \in \arg\min_{\boldsymbol{u}} & \int_t^{t+\Delta t} \|\boldsymbol{u}(\tau)\|^2 \, d\tau \\
& + \lambda \, \Phi_{\mathrm{MPC}}(\boldsymbol{x}(t+\Delta t), \, t+\Delta t) \quad (8) \\
\text{s.t.} \ & \frac{d\boldsymbol{x}(\tau)}{d\tau} = v_\theta(\boldsymbol{x}(\tau), \tau) + \boldsymbol{u}(\tau).
\end{aligned}
$$

The optimisation is performed over $\tau \in [t, t + \Delta t]$, with the state initialised at $\boldsymbol{x}(t) = \boldsymbol{x}_t$. This version of the MPC-Flow framework is described in Algorithm 2. Unlike for MPC-RHC, the choice of intermediate cost $\Phi_{\mathrm{MPC}}$ is non-trivial here, as we are only solving the controlled ODE up to $t + \Delta t$. Simply using $\Phi(\boldsymbol{x})$ would result in a control that would fail to look ahead. We show that the optimal choice for the intermediate loss is the value function itself.

**Theorem 3.2** ($\Delta t$-Horizon optimality). *The $\Delta t$-horizon control defined in* (8) *yields the globally optimal control policy if the terminal cost is chosen as the cost-to-go, i.e.,* $\Phi_{MPC}(\boldsymbol{x}, t + \Delta t) = V(t + \Delta t, \boldsymbol{x})$.

The proof is again a consequence of the principle of optimality and provided in Appendix B.2.2. Since the true value function $V$ is intractable, practical implementations of $\Delta t$-horizon control must rely on approximations. As a general-purpose heuristic, we can estimate the value function using a single Euler step approximation

$$
V(t, \boldsymbol{x}) \approx \Phi(\boldsymbol{x} + (1 - t)v_\theta(\boldsymbol{x}, t)). \quad (9)
$$

This approximates the remaining trajectory as a straight line without control. For an affine linear probability path, the single-step Euler prediction coincides with the Tweedie estimator used in diffusion models, see Proposition 1 of Kim et al. (2025). In diffusion-based settings, approximations of the value function such as (9) are commonly used. For instance, Li et al. (2024) and Uehara et al. (2025) employ such approximations within sequential Monte Carlo sampling schemes (Doucet et al., 2001). An alternative approach is to learn the value function explicitly, for example via soft Q-learning (Haarnoja et al., 2017). However, this requires an additional training phase and is therefore unsuitable for zero-shot applications.

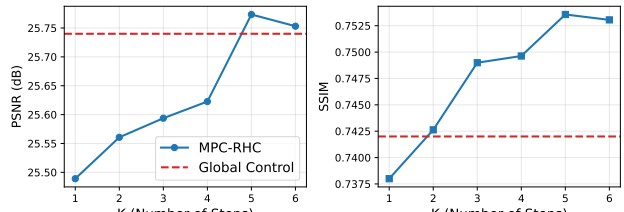

*Figure 3.* Comparison of MPC-RHC with varying $K$ compared to the global optimal control solution. All approaches use the same initial value $\boldsymbol{x}_0$ and $\lambda = 2500$.

## 4. Experiments

In this section we perform experiments exploring the different design choices of MPC-Flow.[1] First, we tackle typical image restoration tasks, both with linear and non-linear degradation operators, on CelebA (Yang et al., 2015). Then, we show that we can scale MPC-RHC with $K = 1$ to the recently released FLUX.2 (Labs, 2025) model and consider style transfer and image colouration.

### 4.1. Computed Tomography

We train a flow-based model on the OrganCMNIST subset of MedMNIST (Yang et al., 2021; 2023), using the flow matching library (Lipman et al., 2024). The flow model is implemented as a UNet with approx. 8 million parameter, which allows us to directly implement the global optimal control problem (6) and compare the solution with MPC-RHC. In Figure 3 we compare the PSNR and SSIM of the final reconstruction for different choices of the discretisation $K$. For an increasing number of steps $K$ we even get a higher PSNR and SSIM than the global control, which can be explained by the fact that the global solution is obtained via a non-convex optimisation problem and may converge to a suboptimal local minimum. We computed the global control using the same data-consistency strength $\lambda$ and used the Adam optimiser (Kingma, 2014) until convergence. Further details and ablations are provided in Appendix D.

### 4.2. Image Restoration

We evaluate MPC-Flow on standard image restoration tasks using the CelebA dataset. All experiments use a pre-trained flow-matching model from Martin et al. (2025), trained with Mini-batch OT Flow (Tong et al., 2024) and a standard Gaussian latent prior. The flow model is implemented as a time-dependent U-Net (Ho et al., 2020). We primarily compare against optimal-control-based methods, specifically the global control approaches FlowGrad (Liu et al., 2023b) and OC-Flow (Wang et al., 2025a). We also include recent flow-based solvers for inverse problems: D-Flow (Ben-Hamu et al., 2024), OT-ODE (Pokle et al., 2024),

---

[1]The code is publicly available at https://github.com/alexdenker/MPCFlow

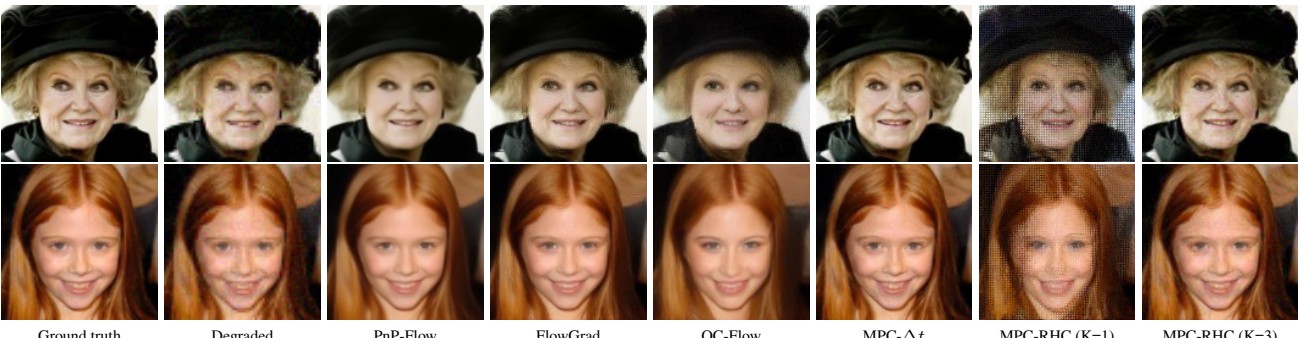

Ground truth    Degraded    PnP-Flow    FlowGrad    OC-Flow    MPC-Δt    MPC-RHC (K=1)    MPC-RHC (K=3)

*Figure 4.* Qualitative results on the CelebA dataset for the image super-resolution task with noise level $\sigma = 0.05$ and $\times 2$ upscaling.

| Method | Denoising $\sigma = 0.2$ | | Deblurring $\sigma = 0.05, \sigma_b = 1.0$ | | Super-res. $\sigma = 0.05, \times 2$ | | Rand. inpaint. $\sigma = 0.01, 70\%$ | | Box inpaint. $\sigma = 0.05, 40 \times 40$ | | Nonl. Debl. $\sigma = 0.05$ | | Time/img [s] |
|---|---|---|---|---|---|---|---|---|---|---|---|---|---|
| | PSNR ↑ / SSIM ↑ | | PSNR / SSIM | | PSNR / SSIM | | PSNR / SSIM | | PSNR / SSIM | | PSNR / SSIM | | |
| Degraded | 20.00 | 0.348 | 27.67 | 0.740 | 10.17 | -0.182 | 11.82 | -0.197 | 22.12 | 0.742 | 21.58 | 0.516 | – |
| OT-ODE | 30.50 | 0.867 | 32.63 | 0.915 | 31.05 | 0.902 | 28.36 | 0.865 | 28.84 | 0.914 | 21.95 | 0.626 | 5.19 |
| D-Flow | 26.42 | 0.651 | 31.07 | 0.877 | 30.75 | 0.866 | 33.07 | 0.938 | 29.70 | 0.893 | **26.41** | 0.682 | 26.31 |
| Flow-Priors | 29.26 | 0.766 | 31.40 | 0.856 | 28.35 | 0.717 | 32.33 | 0.945 | 29.40 | 0.858 | 21.84 | 0.490 | 24.30 |
| PnP-Flow | **32.45** | **0.911** | **34.51** | **0.940** | 31.49 | 0.907 | **33.54** | **0.953** | 30.59 | **0.943** | 22.19 | 0.643 | 5.97 |
| FlowChef | 21.57 | 0.403 | 26.02 | 0.596 | 27.68 | 0.729 | 28.46 | 0.848 | 26.41 | 0.776 | 17.59 | 0.279 | 1.50 |
| FlowDPS | 31.45 | 0.900 | 31.84 | 0.906 | 26.01 | 0.786 | 26.42 | 0.798 | 28.73 | 0.884 | 23.79 | 0.707 | 3.76 |
| OC-Flow | 19.39 | 0.559 | 27.81 | 0.828 | 24.95 | 0.746 | 16.77 | 0.456 | 26.97 | 0.856 | 20.17 | 0.528 | 234.76 |
| FlowGrad | 26.07 | 0.777 | 30.51 | 0.883 | 29.72 | 0.871 | 29.16 | 0.881 | 25.06 | 0.901 | 19.98 | 0.521 | 315.66 |
| MPC-Δt | 31.55 | 0.877 | 33.65 | 0.928 | **31.89** | **0.911** | 32.76 | 0.942 | **30.68** | 0.931 | 25.44 | **0.718** | 89.25 |
| MPC-RHC (K=1) | 27.30 | 0.708 | 32.47 | 0.900 | 18.02 | 0.421 | 18.31 | 0.455 | 26.13 | 0.877 | 17.63 | 0.405 | 2.05 |
| MPC-RHC (K=3) | 29.13 | 0.823 | 33.35 | 0.925 | 31.39 | 0.896 | 33.25 | 0.951 | 30.03 | 0.929 | 21.71 | 0.596 | 175.20 |

*Table 2.* Comparisons of state-of-the-art methods for several inverse problems on CelebA. Results are averaged across 100 test images. Reconstruction time is measured for the **Denoising** task. We compare general flow-based inverse problem solvers (OT-ODE, D-Flow, Flow-Priors PnP-Flow, FlowChef and FlowDPS), optimal control-based methods (FlowGrad, OC-Flow) and our MPC-Flow variants.

Flow-Priors (Zhang et al., 2024), PnP-Flow (Martin et al., 2025), FlowChef (Patel et al., 2025) and FlowDPS (Kim et al., 2025). Results are in Table E1 for linear image restoration tasks, including denoising, deblurring, super-resolution, random inpainting and box inpainting. Further, we provide results for non-linear deblurring. Qualitative results are in Figure 4. Additional results are in Appendix E.

Across most linear restoration tasks, PnP-Flow achieves the highest PSNR and SSIM, with MPC-Δt consistently ranking second. However, for super-resolution MPC-Δt achieves the best performance on both PSNR and SSIM. For non-linear deblurring, PnP-Flow degrades substantially. In this setting, D-Flow yields the highest PSNR, while MPC-Δt achieves the second-highest PSNR and the best SSIM. Across all tasks, MPC-Flow and its variants consistently outperform the global optimal control baselines FlowGrad and OC-Flow, while also achieving lower runtime. For completeness, FlowChef and FlowDPS were included in our evaluation, although both were originally designed for latent diffusion reconstruction tasks and lack explicit regularization strategies, resulting in suboptimal performance. In particular, FlowChef was primarily developed for image editing applications and is not well suited to high-noise im-

age restoration or image inpainting tasks, particularly in settings involving a non-trivial null space, as highlighted in (Erbach et al., 2026). The MPC-RHC variant with $K = 1$ performs poorly for some forward operators; this behaviour is discussed in Appendix E. This configuration corresponds to a direct implementation of Algorithm 3 without additional numerical tricks or heuristics. For each method, we conducted a hyperparameter search on a subset of the CelebA validation dataset and reported the used hyperparams in Appendix E.

### 4.3. Large Vision Models

Many recent large vision models (LVMs), including Stable Diffusion 3.0 (SD.3.0) (Esser et al., 2024) and FLUX.2 (Labs, 2025), are trained using flow-matching objectives and support both text and image conditioning. However, these models provide no guarantees for enforcing task-specific constraints. We apply our approach to two conditional generation tasks with FLUX.2: style transfer and image colouration. Due to the model's scale (32B parameters), we use a 4-bit quantised version to reduce memory usage. Although quantisation is generally suitable for the forward pass, it significantly reduces the accuracy of backpropagation. Our

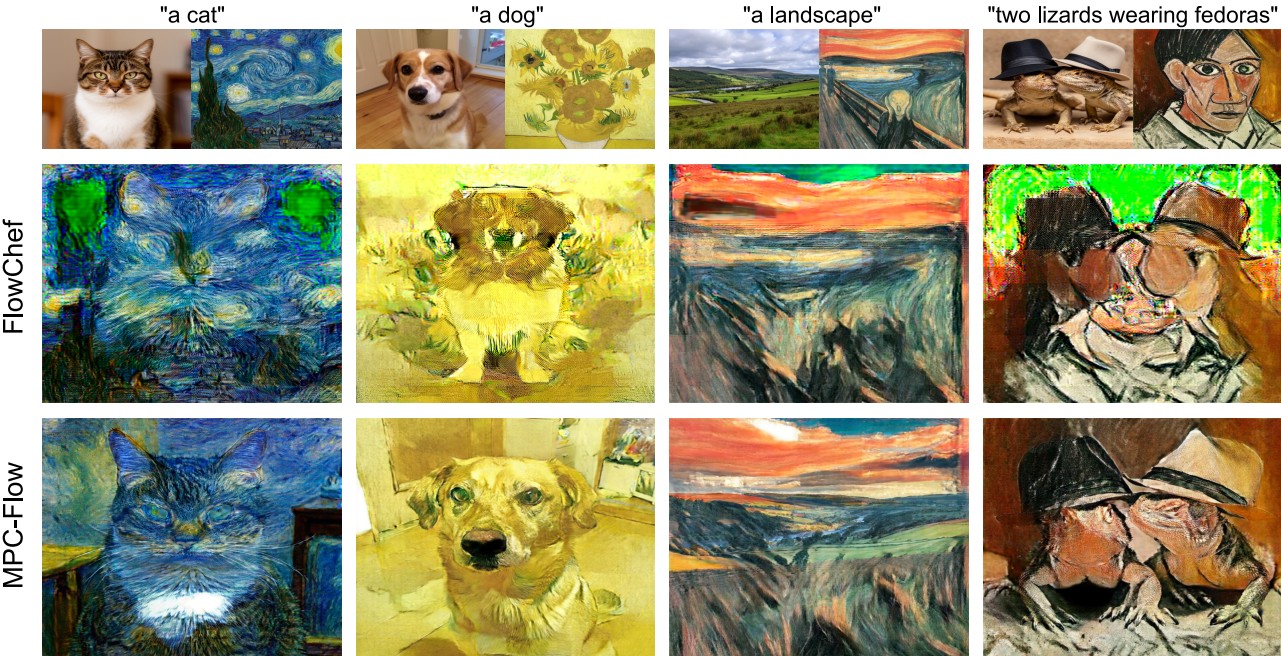

*Figure 5.* Example images generated by FLUX.2 with training-free style transfer guidance by FlowChef (middle row) and MPC (bottom row). The top row shows the image generated without conditioning as well as the prompt used and the reference style image.

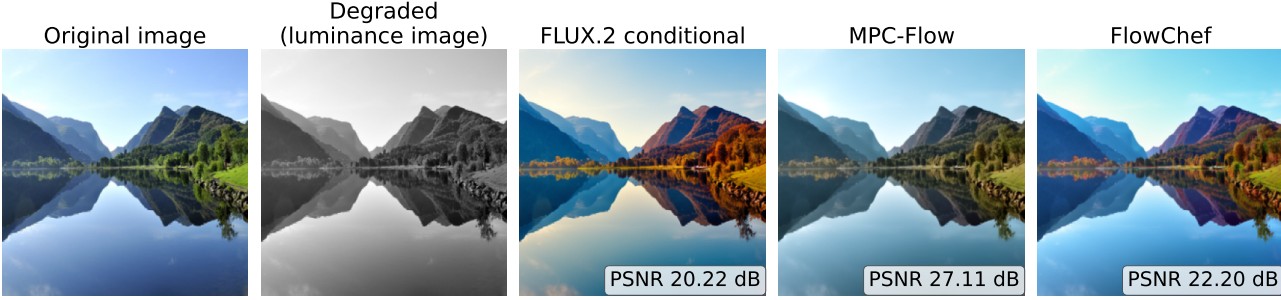

*Figure 6.* Example re-colouration images generated with FLUX.2. Prompt used: "colourize this luminance image".

single-step approach, MPC-RHC ($K = 1$), removes the need for backpropagation through the network, making it particularly suitable for LVM deployments. We compare our method to FlowChef (Patel et al., 2024), which can be viewed as a reparametrisation of our method without control regularisation. All experiments are run on an NVIDIA RTX 3090 GPU with only 24 GB of memory.

**Style transfer** We implemented our approach for style transfer, with the terminal loss

$$\Phi(\boldsymbol{z}) = \big\| \mathrm{Style}(\mathrm{Dec}(\boldsymbol{z})) - \mathrm{Style}(\boldsymbol{x}_{\mathrm{ref}}) \big\|_2^2.$$

where the Style operator is defined via Gram matrices of CLIP ViT-B/16 (Radford et al., 2021) image features and $\boldsymbol{x}_{\mathrm{ref}}$ is the reference style image. Figure 5 shows example images demonstrating qualitatively that on this task our approach delivered more realistic images. Figure 7 shows

different images generated with different hyper-parameters for the same prompt and style image. We fix a budget of 20 conditioning updates per time step. For our method, we vary the balance between terminal loss and control regularisation $\lambda$; for FlowChef, we vary the learning rate on the terminal loss (see Appendix F). As the conditioning strength increases, FlowChef's outputs move closer to the style image, but do so by substantially deviating from the original generation trajectory, which introduces pronounced artefacts. In contrast, our regularised control preserves the generation path more faithfully, allowing the image to adopt stylistic characteristics without severe degradation.

Figure 8 shows a quantitative summary across five prompts and nine style images. As hyperparameters vary, our method exhibits a more favourable trade-off between stylistic similarity and content preservation than FlowChef, consistently

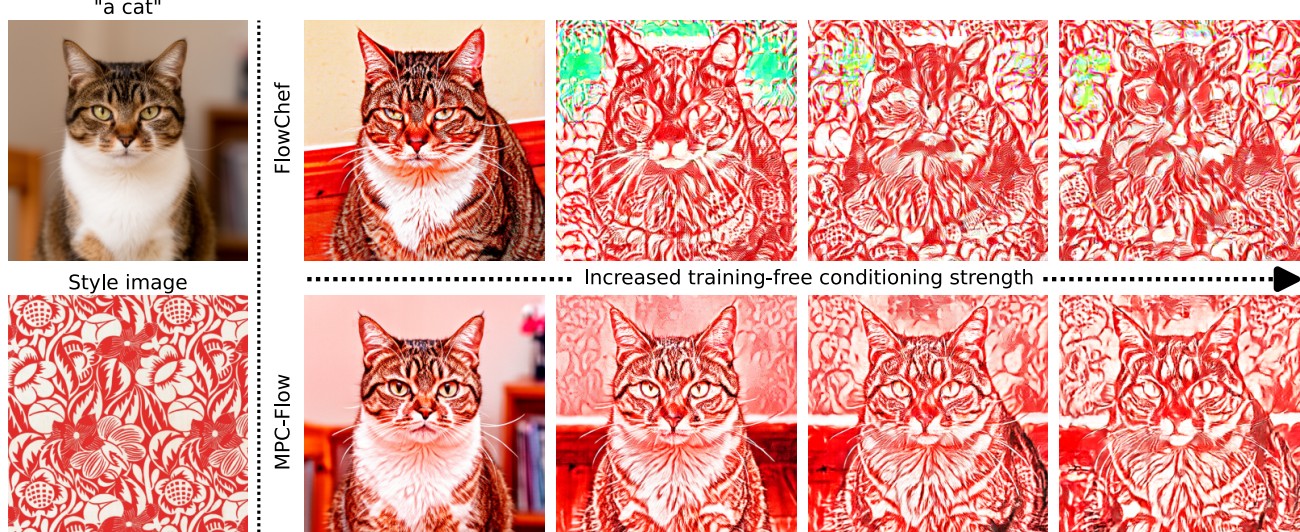

*Figure 7.* Column 1: (Top) An image generated by Flux.2. (Bottom) A reference style image. Remaining columns: Images generated using training-free style transfer with Flux.2 with FlowChef (top) and our method MPC-RHC (K=1) (bottom). From left to right, the methods use increased conditioning on the style reward; the path regularisation introduced by our method enables greater adherence to the original generation path (and hence closer alignment to the original image). For FlowChef, the learning rate of each style optimisation step was varied; for our method, we varied the amount of path regularisation (see Appendix F).

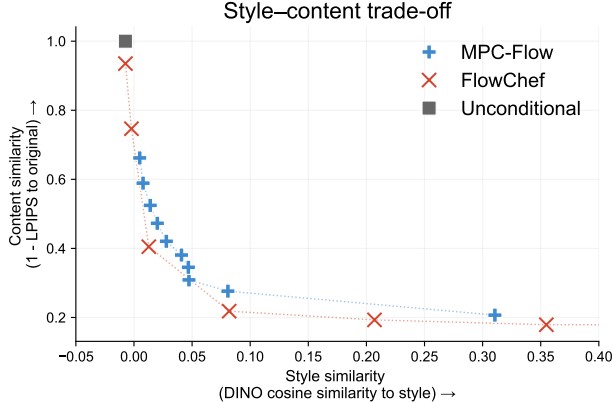

*Figure 8.* Style-content trade-off for MPC-Flow and FlowChef for style transfer with FLUX.2. Metrics recorded over 45 images (5 prompts × 9 style images).

lying on the superior side of the resulting Pareto frontier.

**Image colouration** We also considered the inverse problem of image colouration. Initially, FLUX.2 was prompted to colourise a provided luminance image, but struggled to remain faithful to the degradation process. We addressed this by introducing training-free guidance with MPC-Flow, using the terminal loss

$$\Phi(z) = \frac{1}{HW} \| \boldsymbol{x}_r - \ell(\boldsymbol{x}) \|_2^2$$

where $\boldsymbol{x} = \mathrm{Dec}(\boldsymbol{z}) \in [0,1]^{3 \times H \times W}$ and luminance is defined as $\ell(\boldsymbol{x}) = 0.299\,\boldsymbol{x}_R + 0.587\,\boldsymbol{x}_G + 0.114\,\boldsymbol{x}_B$.

Figure 6 shows example results for image colourisation.

*Table 3.* Results for image colouration with FLUX.2. Hyperparameters for FlowChef and MPC-Flow chosen to maximise PSNR.

| Method | PSNR ↑ | SSIM ↑ |
|---|---|---|
| Unconditional | 18.0 | 0.739 |
| FlowChef | 21.3 | 0.844 |
| MPC-RHC (K=1) | **23.8** | **0.891** |

*Table 4.* Computational cost for image colouration with FLUX.2. Generation used 28 flow steps and 20 conditioning updates/flow step.

| Method | Peak VRAM (GB) | Time (s) |
|---|---|---|
| Unconditional | 17.7 | 53 |
| FlowChef | 18.4 | 92 |
| MPC-RHC (K=1) | 18.4 | 101 |

Qualitatively, this shows that we can steer the flow towards an image that is consistent with the reference observation while remaining close to FLUX.2's learned manifold, resulting in a higher-quality recoloured image. Image similarity metrics averaged over ten images and three random seeds are reported in Table 3. We additionally report the computational cost for this task with FLUX.2 in Table 4, demonstrating the modest overhead of using MPC-Flow (MPC-RHC ($K = 1$)) relative to the unconditional baseline.

**Super-resolution** Lastly, we also implemented MPC-RHC ($K = 1$) for the task of super-resolving a low-resolution image (analogously to the previous image colouration exam-

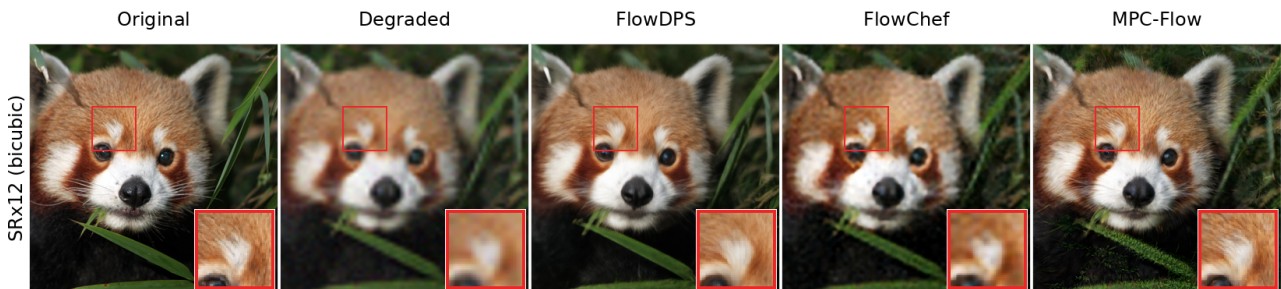

*Figure 9.* Qualitative comparison using SD3.0 on a bicubic $12\times$ SR example.

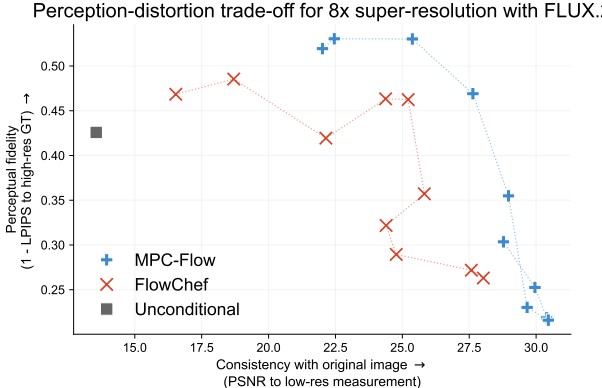

*Figure 10.* Perception-distortion trade-off for 8x super-resolution task with FLUX.2 (metrics averaged over 5 random seeds for a single image recovery task). Corresponding images are shown in Figure F1.

ple). The terminal loss used was a projected measurement-consistency loss in image space,

$$\Phi(\boldsymbol{z}) \;=\; \left\| A^\dagger A \operatorname{Dec}(\boldsymbol{z}) - A^\dagger \boldsymbol{y} \right\|_2^2, \qquad \boldsymbol{y} = A\boldsymbol{x}_{\mathrm{GT}}.$$

where $A$ is the $8\times$ bicubic downsampling operator, $A^\dagger$ is bicubic upsampling back to the high-resolution grid, and $\boldsymbol{y}$ is the low-resolution measurement. Figure F1 shows an improved perception-distortion trade-off with MPC-Flow (MPC-RHC ($K = 1$)) relative to FlowChef and an unconditional baseline.

Additionally, to demonstrate the flexibility of our framework across latent flow models, we follow the noisy inverse-problem evaluation setup of FlowDPS (Kim et al., 2025) and we qualitatively evaluate MPC-Flow ($K = 1$) with SD3.0 (Esser et al., 2024) as the latent flow model. We consider three tasks: $12\times$ super-resolution (SR) from average pooling, $12\times$ super-resolution from bicubic interpolation, and Gaussian deblurring. A comparison for the bicubic $12\times$ SR setting is shown in Figure 9. Further details and visual results are provided in Appendix G, with full comparisons in Figure F3.

## 5. Conclusion & Further Work

We introduce MPC-Flow, a model predictive control framework for conditional generation with flow-based generative models. By decomposing the global trajectory optimisation into a sequence of control subproblems, MPC enables practical and memory-efficient guidance at inference time. We provide formal guarantees connecting MPC-Flow to the underlying optimal control problem. Empirically, MPC-Flow achieves strong performance across linear and non-linear image restoration tasks, while scaling to massive architectures such as FLUX.2 (32B) on consumer hardware. The single-step RHC variant in particular offers a highly efficient and scalable approach to inference-time control.

As a gradient-based method MPC-Flow requires access to a differentiable objective function. Further, the fast MPC-RHC with $K = 1$, does not always achieve good results on some linear inverse problems. MPC-Flow depends on the initial value $\boldsymbol{x}_0$ of the control problem, as discussed in Section H. In all our experiments we simply used a random initialisation for $\boldsymbol{x}_0$. However, more sophisticated initialisations are possible. For example, Ben-Hamu et al. (2024) initialise $\boldsymbol{x}_0$ by first integrating the dynamics backwards in time, starting at $t = 1$ with a naive reconstruction, e.g., computed using the adjoint of the forward operator.

We model the control as a simple additive vector to the flow dynamics. More expressive parametrisations are possible: for example, introducing non-linear control via $\dot{\boldsymbol{x}} = v_\theta(\boldsymbol{x} + \boldsymbol{u}, t)$ (Pandey et al., 2025), or directly adapting the weights of the pre-trained model with $\dot{\boldsymbol{x}} = v_{\theta+\boldsymbol{u}}(\boldsymbol{x}, t)$ using approaches such as low-rank adaptation (Hu et al., 2022) or control networks (Zhang et al., 2023). Further, the theoretical guarantees of Theorem 3.2 and Theorem 3.1 only apply to the continuous-time MPC formulation, while implemented methods introduce approximations: (i) time discretisation of the controlled dynamics via Euler integration; (ii) inexact numerical solution of each MPC subproblem and (iii) for MPC-$\Delta t$, approximation of the value function with a one-step surrogate. However, the discretisation error can be removed by adopting mean-flow models (Geng et al., 2026) instead of standard flow matching approaches.

## Acknowledgments

AD and RB acknowledge support from the EPSRC (EP/V026259/1). AD additionally acknowledges support from DESY (Hamburg, Germany), a member of the Helmholtz Association HGF. GW acknowledges support from the EPSRC CDT in Smart Medical Imaging [EP/S022104/1] and by a GSK Studentship.

## Impact Statement

This work improves inference-time control of flow-based generative models. As with other advances in generative modelling, increased controllability could be misused to generate harmful or misleading synthetic content, including deepfakes. The method does not introduce new generative capabilities beyond existing models and is intended for beneficial applications such as image restoration and scientific imaging.

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

# A. Related Work

**Continuous Normalising Flows**    Continuous Normalising Flows (Chen et al., 2018) trained with Flow Matching (Lipman et al., 2023) have emerged as a powerful methods for generating high-dimensional data, for example for images (Esser et al., 2024) or biological data (Wang et al., 2025b). A key advantage of Flow Matching is its ability to learn Gaussian probability paths corresponding to Optimal Transport (OT) displacements. These paths yield straighter trajectories, enabling faster sampling (Albergo & Vanden-Eijnden, 2023; Liu et al., 2023a) compared to standard diffusion models (Song et al., 2021). However, as training these models remains computationally expensive, there is significant interest in leveraging pre-trained unconditional models as priors for downstream conditional tasks, such as inpainting, super-resolution, or deblurring in imaging inverse problems.

Existing approaches for conditional generation with continuous-time models generally fall into two categories. The first involves conditional training and fine-tuning, where a model is either trained from scratch on paired data (Batzolis et al., 2021) or the weights of the model are adapted (Domingo-Enrich et al., 2025; Zhang et al., 2023). These approaches rely on the availability of additional data and/or require a costly fine-tuning phase. The second category consists of training-free guidance methods, which modify the sampling process to satisfy conditions without altering the unconditional model (Kim et al., 2025; Pokle et al., 2024; Zhang et al., 2024). It is this category that is the subject of this work.

**Guidance for Continuous-time Generative Models**    The problem of steering the generative processes has been heavily studied in the context of diffusion models (Song et al., 2021). This can be framed as Stochastic Optimal Control (SOC), which has been used in the context of inverse problems (Denker et al., 2024; Pandey et al., 2025), fine-tuning (Domingo-Enrich et al., 2025) and model personalisation (Rout et al., 2025). Further, SOC methods have been applied to sampling from unnormalised probability density functions (Zhang & Chen, 2022; Berner et al., 2024; Vargas et al., 2023). These methods rely on stochastic calculus and score-based dynamics, which fundamentally differ from the deterministic ODEs induced by Flow Matching. As an alternative, training-free guidance methods, such as DPS (Chung et al., 2023), ReSample (Song et al., 2024) or DAPS (Zhang et al., 2025), modify the sampling dynamics using approximations of conditional scores. Several of these ideas have been adapted to flow-based models, for example RED-diff (Mardani et al., 2024) was first proposed for diffusion models and later adapted to flow-based models (Erbach et al., 2026). Other approaches are specific to flow-based models; for example, FlowChef (Patel et al., 2024) exploits the deterministic structure of rectified flows to perform gradient-free steering by directly manipulating the vector field during sampling. These methods can be applied at inference time without changing the underlying pre-trained model. However, they typically only provide limited guarantees: the final sample may not satisfy the condition exactly, and the trade-off between consistency to observations and staying close to the original trajectory is only implicit.

**Optimal Control in Flow Models**    Applying optimal control to flow-based models is an active area of research. Existing approaches differ mainly in how gradients are computed, how control cost is handled, and the associated trade-offs of memory and computation time. To mitigate the memory cost associated with large scale optimisation problems, several works utilise adjoint sensitivity methods (Chen et al., 2018). FlowGrad (Liu et al., 2023b) leverages a time-discrete version of the adjoint ODE to compute control gradients. To reduce the computational cost, they introduce a metric for ODE straightness, allowing for fewer backward time steps.

However, this approach still requires solving the full forward ODE at each optimisation step, and notably, the running cost of the control is excluded from the gradient computation. Similarly, OC-Flow (Wang et al., 2025a) employs a time-discrete adjoint for gradient computation but implements a control regularisation implicitly via weight decay.

Both FlowGrad and OC-Flow focus primarily on image editing tasks. D-Flow (Ben-Hamu et al., 2024) only optimises the initial value of the ODE, without introducing additional control terms. Unlike adjoint-based approaches, D-Flow directly backpropagates gradients through the discrete ODE solver. While this avoids adjoint approximation errors, it drastically increases the memory cost as the full trajectory and all model evaluations have to be kept in memory. Finally, VGG-Flow (Liu et al., 2025) estimates the value function gradient and fine-tunes the underlying flow-based model via a value gradient matching loss. From these above-mentioned models only D-Flow has been applied to inverse problems.

Concurrently, HardFlow (Li et al., 2025) introduces a training-free framework for hard-constrained sampling based on receding-horizon model predictive control, closely related to the approach proposed in this work.

# B. Proofs

## B.1. Existence of Optimal Control

Following the standard literature, see e.g. (Fleming & Rishel, 2012), we give an existence proof of the optimal control for our setting.

**Proposition B.1** (Existence of optimal control). *Under the assumptions that*

1) $v_\theta$ *is continuous in both arguments*

2) $v_\theta$ *is uniformly Lipschitz in t, i.e., $\|v_\theta(\boldsymbol{x}_1, t) - v_\theta(\boldsymbol{x}_2, t)\| \le L\|\boldsymbol{x}_1 - \boldsymbol{x}_2\|$ for some $L > 0$,*

3) $v_\theta$ *has at most linear growth, i.e., $\|v_\theta(\boldsymbol{x}, t)\| \le a(t) + b\|\boldsymbol{x}\|$ with $a \in L^1([0, 1])$ and $b > 0$,*

4) $\Phi : \mathbb{R}^n \to \mathbb{R}$ *is lower semicontinuous,*

*there exists an optimal control $\boldsymbol{u}^* \in L^2([0, 1])$ of the control problem*

$$
\begin{aligned}
\min_{\boldsymbol{u}} &\left\{ \mathcal{J}(\boldsymbol{u}) := \int_0^1 \|\boldsymbol{u}(t)\|^2 dt + \Phi(\boldsymbol{x}(1)) \right\} \\
s.t. \quad & d\boldsymbol{x}(t)/dt = v_\theta(\boldsymbol{x}(t), t) + \boldsymbol{u}(t), \\
& t \in [0, 1], \boldsymbol{x}(0) = \boldsymbol{x}_0.
\end{aligned}
\tag{10}
$$

*Proof.* First, we observe that the infimum is finite, i.e., $\inf \mathcal{J}(\boldsymbol{u}) < \infty$, as $\boldsymbol{u} = 0$ is an admissible control and we have the grwoth condition on $v_\theta$. Let $(\boldsymbol{u}_k)$ be a minimising sequence such that $\mathcal{J}(\boldsymbol{u}_k) \to \inf \mathcal{J}(u)$. As the control cost $\int_0^1 \|\boldsymbol{u}_k(t)\|^2 dt$ is bounded, we have a weakly convergent subsequence (which we still denote it as $\boldsymbol{u}_k$ for an easier notation), such that $\boldsymbol{u}_k \rightharpoonup \boldsymbol{u}^*$ in $L^2([0, 1])$. We have to show that the optimal candidate $\boldsymbol{u}^*$ is actually a minimiser, i.e., $\mathcal{J}(\boldsymbol{u}^*) = \inf \mathcal{J}(u)$.

For this, we first look at the convergence of the trajectories. Let $\boldsymbol{x}_k$ and $\boldsymbol{x}^*$ denote the trajectories corresponding to $\boldsymbol{u}_k$ and $\boldsymbol{u}^*$ respectively. We can express the trajectories as

$$
\boldsymbol{x}_k(t) = \boldsymbol{x}_0 + \int_0^t v_\theta(\boldsymbol{x}_k(s), s) ds + \int_0^t \boldsymbol{u}_k(s) ds, \quad \boldsymbol{x}^*(t) = \boldsymbol{x}_0 + \int_0^t v_\theta(\boldsymbol{x}^*(s), s) ds + \int_0^t \boldsymbol{u}^*(s) ds.
\tag{11}
$$

Let us define the integrated control error as $E_k(t) := \int_0^t (\boldsymbol{u}_k(s) - \boldsymbol{u}^*(s)) ds$. Since $u_k \rightharpoonup u^*$ weakly in $L^2([0, 1])$, we obtain $E_k(t) \to 0$ for any fixed $t$. Furthermore, as the controls $(\boldsymbol{u}_k)$ are bounded, we have that $E_k(t)$ is bounded. By the Arzelà-Ascoli, there is then a subsequence of $E_k$ which converges uniformly to 0, i.e., there is $\epsilon_k := \sup_{t \in [0,1]} \|E_k\| \to 0$.

Further, we get for the trajectories

$$
\|\boldsymbol{x}_k(t) - \boldsymbol{x}^*(t)\| \le \int_0^t \|v_\theta(\boldsymbol{x}_k(s), s) - v_\theta(\boldsymbol{x}^*(s), s)\| ds + \|E_k(t)\|
\tag{12}
$$

$$
\le \int_0^1 L\|\boldsymbol{x}_k(s) - \boldsymbol{x}^*\| + \epsilon_k
\tag{13}
$$

$$
\le \epsilon_k \exp(Lt)
\tag{14}
$$

where the last inequality is due to Grönwall. As $\epsilon_k \to 0$, we have that $\boldsymbol{x}_k(t) \to \boldsymbol{x}^*(t)$ uniformly on $[0, 1]$. In particular, $\boldsymbol{x}_k(1) \to \boldsymbol{x}^*(1)$.

Finally, we get

$$
\mathcal{J}(\boldsymbol{u}^*) = \int_0^1 \|\boldsymbol{u}^*(t)\|^2 dt + \Phi(\boldsymbol{x}^*(1)) \le \liminf_{k \to \infty} \left( \int_0^1 \|\boldsymbol{u}_k(t)\|^2 dt + \Phi(\boldsymbol{x}_k(1)) \right) = \inf \mathcal{J}(\boldsymbol{u}),
\tag{15}
$$

as both the squared $L^2$ norm and $\Phi$ are lower semicontinuous. Thus, $\boldsymbol{u}^*$ is an optimal control. $\square$

Importantly, we do not need that the control is in some compact set due to the control cost $\int_0^1 \|u(t)\| dt$.

### B.2. Optimality of MPC

B.2.1. PROOF OF THEOREM 3.1

*Proof.* This result follows directly from Bellman's principle of optimality, see e.g. Liberzon (2011). Let the optimal trajectory be $\boldsymbol{x}^*$. Suppose we are at time $t$ in state $\boldsymbol{x}^*(t)$. The global cost can be split

$$\mathcal{J}(\boldsymbol{u}) = \int_0^t \|\boldsymbol{u}\|^2 d\tau + \int_t^1 \|\boldsymbol{u}\|^2 d\tau + \lambda\Phi(\boldsymbol{x}(1)).$$

Since the past cost in $[0,t]$ is fixed, minimising the global cost is equivalent to minimising the future cost, i.e. the second and third terms. The solution to the sub-problem (7) at time $t$ is exactly the restriction of the global optimal control to the interval $[t,1]$. Thus, re-optimising at each step yields the same control as (5). □

B.2.2. PROOF OF THEOREM 3.2

*Proof.* The value function $V(t, \boldsymbol{x}(t))$ can be expressed as

$$\min_{\boldsymbol{u}} \left\{ \int_t^{t+\Delta t} \|\boldsymbol{u}(\tau)\|^2 d\tau + V(t+\Delta t, \boldsymbol{x}(t+\Delta t)) \right\}, \tag{16}$$

for a time step $\Delta t$. This optimisation problem is identical to (8) with $\Phi_{\text{MPC}}(\boldsymbol{x}(t + \Delta t)) = V(t + \Delta t, \boldsymbol{x}(t + \Delta t))$. This means that minimising (8) for this choice of terminal cost evaluates the value function at every time step and thus we obtain the optimal control. □

## C. Validating optimality guarantees with a toy example

In Theorem 3.1, we establish the optimality of the receding-horizon control (RHC) strategy under the assumption that each finite-horizon sub-problem is solved exactly. We complement this theoretical result with an empirical validation on a simple two-dimensional controlled flow-matching example, where the flow trajectories can be visualised directly.

We train a flow model using flow matching to generate samples lying on the boundary of a regular hexagon with side length 2. Training pairs are constructed by sampling $\boldsymbol{x}_0 \sim \mathcal{N}(0, I)$ and $\boldsymbol{x}_1$ uniformly from the hexagon boundary, and defining an affine interpolation path between $\boldsymbol{x}_0$ and $\boldsymbol{x}_1$. The model then learns a time-dependent velocity field $\boldsymbol{v}(\boldsymbol{x}_t, t)$ that matches the instantaneous velocity of this affine path at each time step (see equation (4)). Integrating $\boldsymbol{v}$ thus transports isotropic noise toward the hexagon boundary along approximately straight-line trajectories.

At inference time, we apply MPC to steer the generative trajectory toward a specific mode of the distribution. In particular, the terminal loss is chosen as

$$\Phi(\boldsymbol{x}) = \|\boldsymbol{x} - \boldsymbol{x}_{\text{corner}}\|_2^2,$$

where $\boldsymbol{x}_{\text{corner}}$ denotes the coordinates of the lower-right corner of the hexagon. While the learned flow alone produces samples uniformly along the hexagon's boundary, the MPC controller biases the otherwise affine transport path toward the selected corner, providing a concrete and interpretable setting in which to assess the optimality guarantees of Theorem 3.1.

**MPC-RHC** We first present results for MPC-RHC in Figure C1. Recall that at time step $t$, MPC-RHC plans the remaining trajectory as an Euler discretisation of $\boldsymbol{x}_{[t,1]}$ with $K$ steps. It may be observed that the MPC-RHC methods approximate the globally optimal control as $K$ increases. Further, all methods achieve the desired target point while balancing control energy and the terminal loss. Figure C2a demonstrates empirically that the error to the globally optimal control decreases as $K$ increases, while Figure C2b demonstrates that as $K$ increases, the method finds a stable balance between control regularisation and satisfying the terminal objective.

**MPC-$\Delta t$** For MPC-$\Delta t$ we use a single Euler step approximation for the value functions, see (9). For this 2D example, we can numerically solve the Hamilton–Jacobi–Bellman (HJB) equation using a semi-Lagrangian scheme that propagates the value function. This yields both the value function and the optimal control. In Figure C3 we compare this true value function against the one step approximation. At early times ($t \approx 0$) the discrepancy is large, but is decreases as time progresses. Due to this discrepancy at small times we do not recover the global optimal control. However, as time increases the MPC-$\Delta t$ control follows the global control, see also (f) in Figure C3.

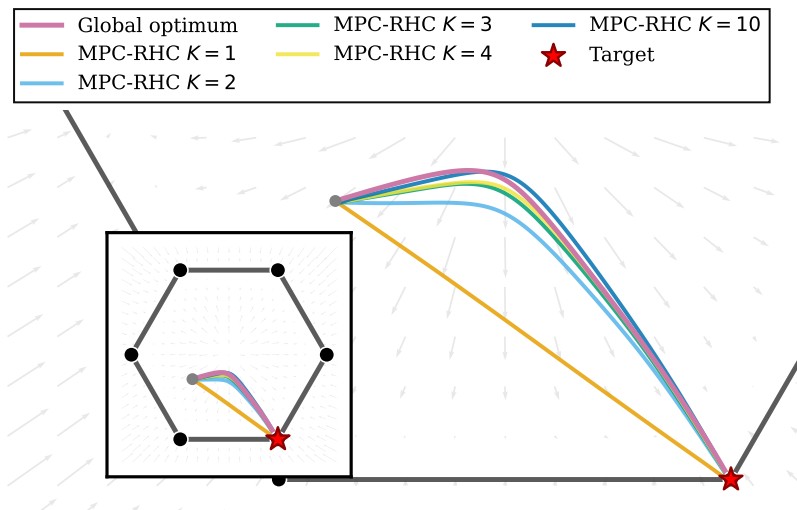

*Figure C1.* MPC-RHC for the 2D hexagon toy example. The flow model is trained to transport random points to the hexagon boundary. MPC-RHC's terminal loss is $\Phi(\boldsymbol{x}) = \|\boldsymbol{x} - \boldsymbol{x}_{\text{corner}}\|^2$, where $\boldsymbol{x}_{\text{corner}}$ are the coordinates of the lower right corner of the hexagon. The "global optimum" path represents the exact solution to the optimal control problem (solved globally over the whole trajectory $\boldsymbol{x}_{[0,1]}$).

## D. Computed Tomography

We make use of the OrganCMNIST subset of MedMNIST (Yang et al., 2021; 2023) to provide ablations on the hyperparameters of the different variations of MPC-Flow. The OrganCMNIST dataset contains $64 \times 64$px adnominal CT images. In total there are $23\,582$ images, split into $12\,975$ training, $2392$ validation and $8216$ test images. We train an unconditional flow model on this dataset. Here, we make use of the flow matching library (Lipman et al., 2024). The flow model is implemented as a time-dependent UNet with approx. 8Mio parameters (Dhariwal & Nichol, 2021). For the forward operator we make use of the parallel-beam Radon transform with $18$ angles and corrupt the simulated measurements with $1\%$ relative additive Gaussian noise.

**MPC-$\Delta t$**  We first provide ablations for MPC-$\Delta t$, see Algorithm 2. For every $t'$, we have to solve the optimisation problem

$$\boldsymbol{u}^* = \arg\min_{\boldsymbol{u}} \left[ \|\boldsymbol{u}\|^2 + \lambda \Phi_{\text{MPC}}(\boldsymbol{x}_{t'+\Delta t},\, t' + \Delta t) \right]$$

$$\text{with } \boldsymbol{x}_{t'+\Delta t} = \boldsymbol{x}_{t'} + \Delta t \big( v_\theta(\boldsymbol{x}_{t'}, t') + \boldsymbol{u} \big),$$

with $\lambda = \tilde{\lambda}/\Delta t$, where the intermediate loss is defined as

$$\Phi_{\text{MPC}}(\boldsymbol{x}_{t'+\Delta t}, t' + \Delta t) \coloneqq \Phi(\boldsymbol{x}_{t'+\Delta t} + (1 - (t' + \Delta t)) v_\theta(\boldsymbol{x}_{t'+\Delta t}, t' + \Delta t)).$$

We solve this optimisation problem using the Adam optimiser (Kingma, 2014). For these experiments, we use a fixed learning rate of $0.1$ and re-initialise the optimiser with $\boldsymbol{u}_{\text{init}} = \boldsymbol{0}$ at every new time step. We vary both the data-consistency strength $\lambda$ and the step size $\Delta t = 1/N$. In Figure D1a we study the effect of the data-consistency strength $\lambda$, while keeping $N = 40$ constant. Here, we observe that the optimal value, both for the PSNR and SSIM, is at $\lambda \approx 1 \times 10^4$ and then degrades for larger values. Further, for a fixed $\lambda = 1 \times 10^4$ we show the dependency on the discretisation $N$ in Figure D1b. For a coarser discretisation, the flow model has less ability to explore fine-grained modes of the image distribution, and returns a smoother image; PSNR favours smoother images over incorrectly detailed images, so PSNR rises as $N$ decreases in this case. We show exemplary reconstruction in Figure D2.

## E. Image Restoration

This section complements the experimental study on linear imaging tasks (denoising, deblurring, super-resolution, random inpainting, and box inpainting) and a nonlinear imaging task (nonlinear deblurring).

We report qualitative results for the best-performing methods (or those most suitable for comparison with our proposed approach) across each image restoration task; see Figures E1 to E6 for results on the CelebA test set.

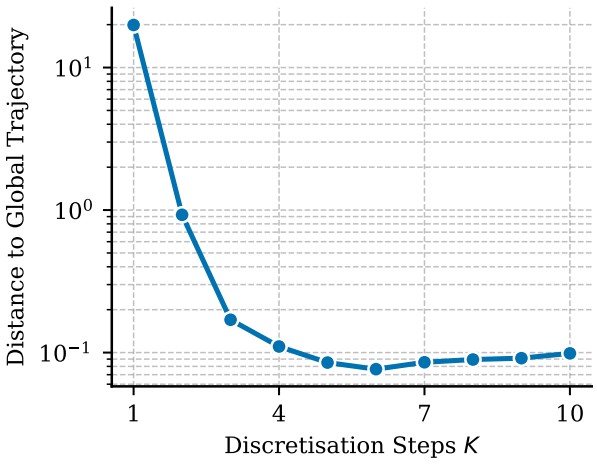

(a) Distance to the globally optimal trajectory for different discretisation steps $K$.

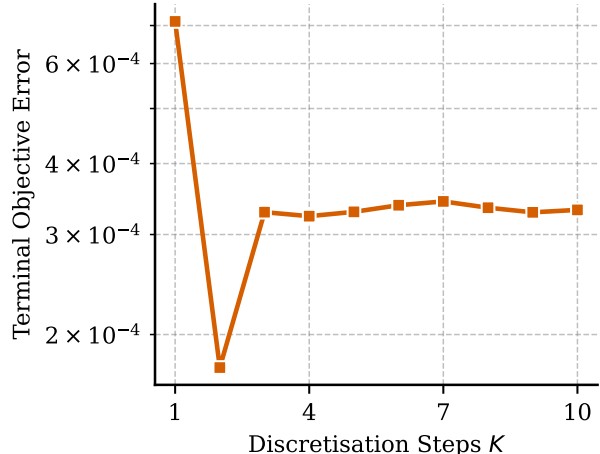

(b) Terminal error of MPC-RHC trajectories for different discretisation steps $K$.

*Figure C2.* Effect of discretisation steps $K$ on MPC-RHC performance.

We also report the hyperparameters used in each experiment. Hyperparameters were selected on a subset of the CelebA validation split. Table E2 summarises the hyperparameters for FlowGrad, OC-Flow, FlowChef and FlowDPS, along with our proposed methods.

For the linear restoration tasks, the hyperparameters of OT-ODE, D-Flow, Flow-Priors, and PnP-Flow are taken from Martin et al. (2025). We use 100 time steps for all tasks. For the nonlinear task, hyperparameters were selected on a subset of the validation split.

Note that for the nonlinear deblurring task (see Table E1), we adopt the nonlinear forward operator used in Chung et al. (2023); Denker et al. (2024); Wang et al. (2024). The operator is parameterised by a trained neural network (Tran et al., 2021), inducing a highly nonlinear blur. Our implementation follows the publicly available codebase of Blur-Kernel-Space-Exploring[2].

| Method | Denoising $\sigma = 0.2$ | | | Deblurring $\sigma = 0.05$, $\sigma_b = 1.0$ | | | Super-res. $\sigma = 0.05$, $\times 2$ | | | Rand. inpaint. $\sigma = 0.01$, 70% | | | Box inpaint. $\sigma = 0.05$, $40 \times 40$ | | | Nonl. Debl. $\sigma = 0.05$ | | | Time/img [s] |
|---|---|---|---|---|---|---|---|---|---|---|---|---|---|---|---|---|---|---|---|
| | PSNR $\uparrow$ / SSIM $\uparrow$ / LPIPS $\downarrow$ | | | PSNR / SSIM / LPIPS | | | PSNR / SSIM / LPIPS | | | PSNR / SSIM / LPIPS | | | PSNR / SSIM / LPIPS | | | PSNR / SSIM / LPIPS | | | |
| Degraded | 20.00 | 0.348 | 0.373 | 27.67 | 0.740 | 0.125 | 10.17 | 0.182 | 0.827 | 11.82 | 0.197 | 1.034 | 22.12 | 0.742 | 0.213 | 21.58 | 0.516 | 0.370 | – |
| OT-ODE | 30.50 | 0.867 | 0.110 | 32.63 | 0.915 | 0.096 | 31.05 | 0.902 | 0.091 | 28.36 | 0.865 | 0.137 | 28.84 | 0.914 | 0.102 | 21.95 | 0.626 | 0.118 | 5.19 |
| D-Flow | 26.42 | 0.651 | 0.078 | 31.07 | 0.877 | 0.053 | 30.75 | 0.866 | 0.027 | 33.07 | 0.938 | 0.021 | 29.70 | 0.893 | 0.035 | **26.41** | 0.682 | 0.069 | 26.31 |
| Flow-Priors | 29.26 | 0.766 | 0.137 | 31.40 | 0.856 | 0.056 | 28.35 | 0.717 | 0.101 | 32.33 | 0.945 | 0.018 | 21.84 | 0.490 | 0.318 | 21.84 | 0.490 | 0.318 | 24.30 |
| PnP-Flow | **32.45** | **0.911** | 0.056 | **34.51** | **0.940** | 0.046 | 31.49 | 0.907 | 0.055 | **33.54** | **0.953** | 0.021 | 30.59 | 0.943 | 0.043 | 22.19 | 0.643 | 0.263 | 5.97 |
| FlowChef | 21.57 | 0.403 | 0.301 | 26.02 | 0.596 | 0.208 | 27.68 | 0.729 | 0.085 | 28.46 | 0.848 | 0.061 | 26.41 | 0.776 | 0.067 | 17.59 | 0.279 | 0.451 | 1.50 |
| FlowDPS | 31.45 | 0.900 | 0.041 | 31.84 | 0.906 | 0.043 | 26.01 | 0.786 | 0.118 | 26.42 | 0.798 | 0.108 | 28.73 | 0.884 | 0.063 | 23.79 | 0.707 | 0.157 | 3.76 |
| OC-Flow | 19.39 | 0.559 | 0.207 | 27.81 | 0.828 | 0.082 | 24.95 | 0.746 | 0.114 | 16.77 | 0.456 | 0.418 | 26.97 | 0.856 | 0.075 | 20.17 | 0.528 | 0.170 | 234.76 |
| FlowGrad | 26.07 | 0.777 | 0.116 | 30.51 | 0.883 | 0.067 | 29.72 | 0.871 | 0.050 | 29.16 | 0.881 | 0.039 | 25.06 | 0.901 | 0.068 | 19.98 | 0.521 | 0.170 | 315.66 |
| MPC-$\Delta t$ | 31.55 | 0.877 | **0.029** | 33.65 | 0.928 | **0.017** | **31.89** | **0.911** | **0.018** | 32.76 | 0.942 | 0.015 | 30.68 | 0.931 | **0.021** | 25.44 | **0.718** | **0.067** | 89.25 |
| MPC-RHC (K=1) | 27.30 | 0.708 | 0.092 | 32.47 | 0.900 | 0.030 | 18.02 | 0.421 | 0.325 | 18.31 | 0.455 | 0.419 | 26.13 | 0.877 | 0.046 | 17.63 | 0.405 | 0.244 | 2.05 |
| MPC-RHC (K=3) | 29.13 | 0.823 | 0.066 | 33.35 | 0.925 | 0.023 | 31.39 | 0.896 | 0.029 | 33.25 | 0.951 | **0.013** | 30.03 | 0.929 | 0.026 | 21.71 | 0.596 | 0.133 | 175.20 |

*Table E1.* Comparisons of state-of-the-art methods for several inverse problems on CelebA. Results are averaged across 100 test images. Reconstruction time is measured for the **Denoising** task. We compare general flow-based inverse problem solvers (OT-ODE, D-Flow, Flow-Priors PnP-Flow, FlowChef and FlowDPS), optimal control-based methods (FlowGrad, OC-Flow) and our MPC-Flow variants.

### E.1. MPC-RHC $K = 1$ for Linear Inverse Problems

A limitation of the single-step approach emerges when applied to linear inverse problems where the forward operator $\mathbf{A}$ has a large null space. Typical examples include inpainting or certain instances of super-resolution, where some pixels are completely unobserved. Consider the case with a quadratic terminal loss $\Phi(\boldsymbol{x}) = \frac{1}{2}\|\mathbf{A}\boldsymbol{x} - \boldsymbol{y}\|_2^2$. The MPC-RHC $K = 1$

[2]https://github.com/VinAIResearch/blur-kernel-space-exploring

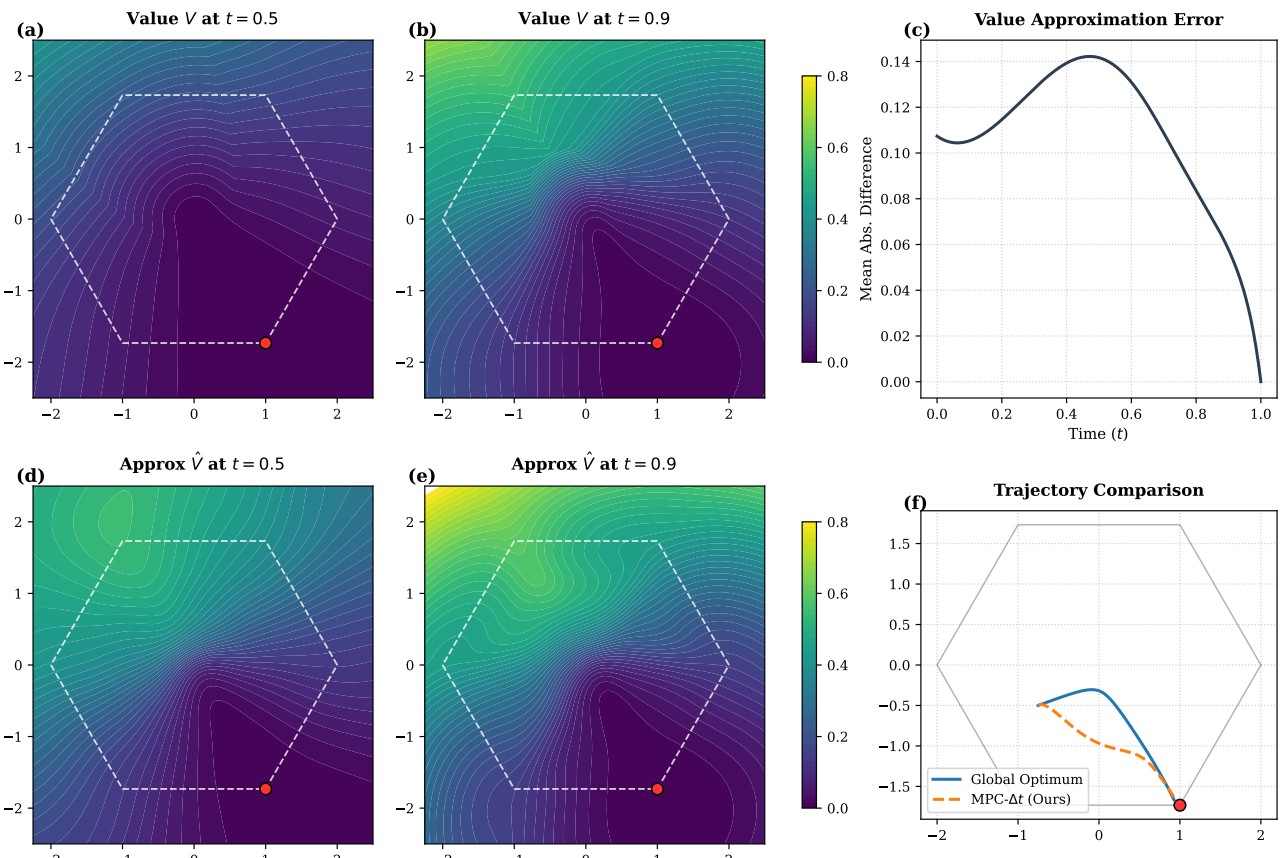

*Figure C3.* MPC-$\Delta t$ for the 2D hexagon toy example. In (a) and (b) we show the value function (computed using the HJB) at $t = 0.5$ and $t = 0.9$. Similarly, in (d) and (e) we show the respective one-step approximation. The approximation error over time is in (c). Finally, (f) shows the global optimal control and the trajectory computed with MPC-$\Delta t$.

objective function at time $t'$ then reduces to

$$\min_{\boldsymbol{u} \in \mathbb{R}^d} \left\{ \mathcal{J}_{K=1}(\boldsymbol{u}) = (1 - t')\|\boldsymbol{u}\|^2 + \frac{\lambda}{2}\|\mathbf{A}\boldsymbol{x}' - \boldsymbol{y}\|^2 \right\}, \quad \boldsymbol{x}' = \boldsymbol{x} + (1 - t')\big(v_\theta(\boldsymbol{x}, t') + \boldsymbol{u}\big), \tag{17}$$

and the gradient w.r.t. the control $\boldsymbol{u}$ is given as

$$\nabla_{\boldsymbol{u}} \mathcal{J}_{K=1}(\boldsymbol{u}) = (1 - t')\big[2\boldsymbol{u} + \lambda\mathbf{A}^T(\mathbf{A}\boldsymbol{x}' - \boldsymbol{y})\big] \tag{18}$$

In particular, the second term of the gradient is orthogonal to the null space of $\mathbf{A}$. This means that in directions of the null space the gradient only depends on the quadratic regularisation term. As we initialise the control with zero, components of $\boldsymbol{u}$ corresponding to the null space of $\mathbf{A}$ remain zero. Consider the super-resolution tasks. Here, super-resolution is implemented as keeping only every second pixel. In Figure 4 the null space of the operator, corresponding to a pixel grid, is still visible in the MPC-RHC $K = 1$ reconstruction and leads to artifacts. For these applications, the control is seemingly changing the observed pixels too fast and the flow model is not able to properly fill in the missing information.

## F. Image generation with Flux.2

**Style loss explicit definition** We define

$$\Phi(\boldsymbol{z}) = \big\|\text{Style}(\text{Dec}(\boldsymbol{z})) - \text{Style}(\boldsymbol{x}_{\text{ref}})\big\|_2^2,$$

where the Style operator uses CLIP ViT-B/16 (Radford et al., 2021) image features. Let $F(\boldsymbol{x}) \in \mathbb{R}^{N \times d}$ denote the patch features extracted from the image encoder, and define the Gram matrix $G(\boldsymbol{x}) = F(\boldsymbol{x})^\top F(\boldsymbol{x})$. We then set $\text{Style}(\boldsymbol{x}) = G(\boldsymbol{x})$ and take $\boldsymbol{x}_{\text{ref}}$ to be the reference style image.

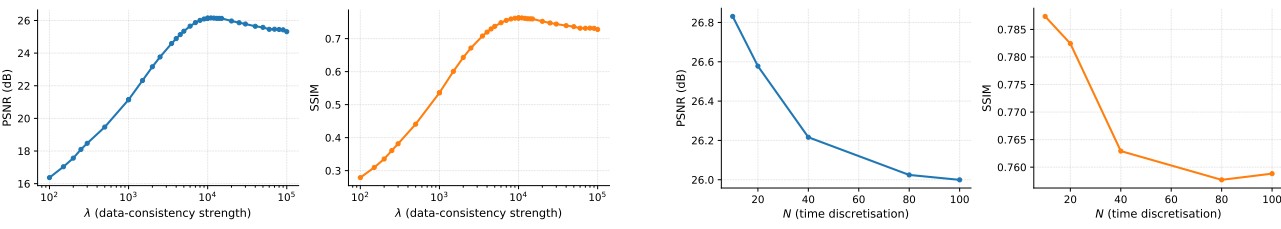

*(a)* MPC-$\Delta t$: Dependence on data-consistency strength $\lambda$.   *(b)* MPC-$\Delta t$: Dependence on discretisation $N$.

*Figure D1.* Effect of $\Delta t = 1/N$ and the data-consistency strength $\lambda$ on the performance of MPC-$\Delta t$. We plot both the mean PSNR and SSIM over 10 images from the test set.

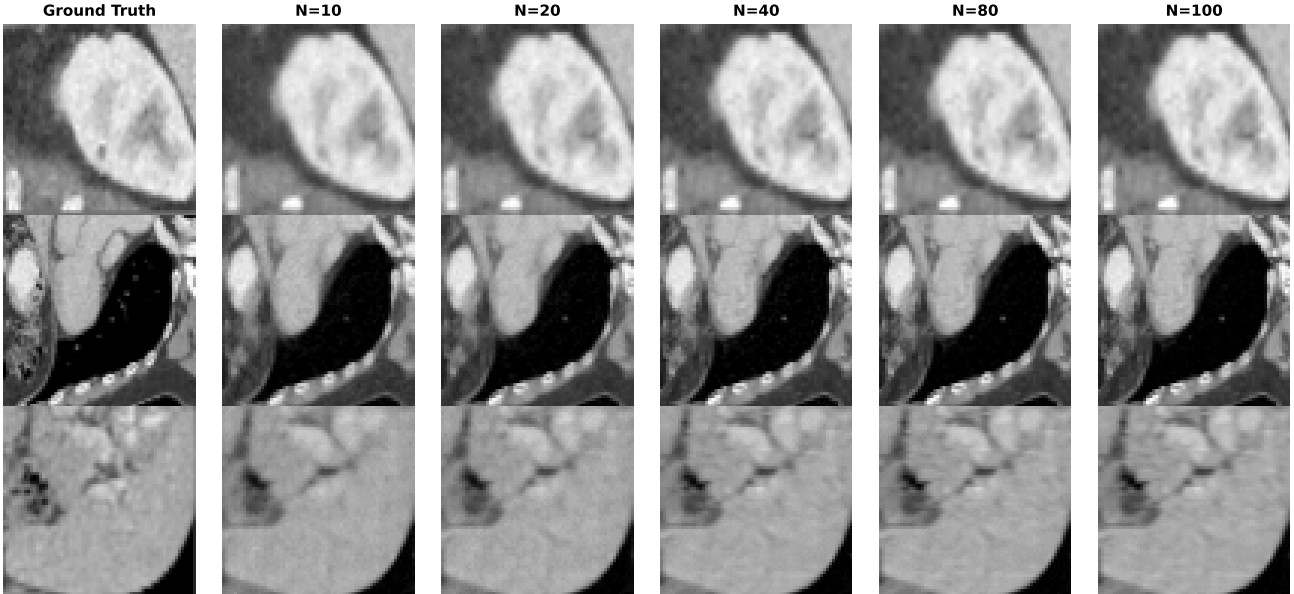

*Figure D2.* Example reconstruction using MPC-$\Delta t$ with $\lambda = 1 \times 10^4$ with different choices of the discretisation resolution $N = 10, 20, 40, 80, 100$. For lower $N$, the images are slightly smoother, giving rise to a higher PSNR in Figure D1b.

**Super-resolution**   Figure F1 shows example images for the 8x super-resolution task using MPC-Flow and FlowChef with FLUX.2.

**Hyperparameter settings**   We used FLUX.2's default image-generation settings: 28 denoising steps and a guidance scale of 4. For FlowChef, we implemented Algorithm 1 from Patel et al. (2024).

For both FlowChef and MPC-Flow, we fixed a budget of 20 inference steps per denoising step. In FlowChef, the strength of conditioning was controlled via the learning rate on the conditioning objective, which was varied from $1 \times 10^{-4}$ to $5 \times 10^{-1}$. In MPC-Flow, rather than scaling the terminal objective directly, we rewrote the control objective

$$\int_t^1 \|\boldsymbol{u}(\tau)\|_2^2 \, d\tau + \lambda \, \Phi(\boldsymbol{x}(1))$$

into the equivalent form

$$\int_t^1 \rho \, \|\boldsymbol{u}(\tau)\|_2^2 \, d\tau + \Phi(\boldsymbol{x}(1)).$$

This reparameterisation isolates the trade-off between control regularisation and terminal quality into a single path-regularisation parameter $\rho$, yielding a numerically more stable tuning knob.

The choice of learning rate and $\rho$ depended on the form and scale of the terminal cost $\Phi$. In the style transfer experiments, $\Phi$ was defined as a perceptual loss computed through a neural network, whereas in the luminance (recoloring) experiments, $\Phi$

| Method | Image restoration tasks | | | | | |
|---|---|---|---|---|---|---|
| | Denoising | Deblurring | Super-res. | Rand. inpaint. | Box inpaint. | Nonl. debl. |
| FlowChef | (0.01,1) | (0.01,1) | (0.1,100) | (1.0,100) | (0.01,10) | (0.01,10) |
| FlowDPS | (0.1,100) | (0.5,75) | (0.1,100) | (0.1,100) | (0.1,100) | (0.01,100) |
| FlowGrad | (0.01,1.5,80) | (0.01,1.5,80) | (0.01,1.5,80) | (0.01,1.0,80) | (0.01,1.0,80) | (0.01,0.5,100) |
| OC-Flow | (0.01,0.5,60,0.995) | (0.01,0.5,60,0.995) | (0.01,0.5,60,0.995) | (0.01,0.5,60,0.995) | (0.01,0.5,60,0.995) | (0.01,0.5,80,0.995) |
| MPC-$\Delta t$ | (15,20,0.1) | (7.5,20,0.1) | (7.5,20,0.1) | (0.5,20,0.1) | (5,20,0.1) | (20,20,0.1) |
| MPC-RHC ($K=1$) | (0.1,20,0.1) | (0.063,20,0.1) | (1,20,0.1) | (1,20,0.1) | (0.040,20,0.1) | (0.063,20,0.1) |
| MPC-RHC ($K=3$) | (0.1,20,0.05) | (0.063,20,0.05) | (0.063,20,0.05) | (1,20,0.05) | (1,20,0.05) | (0.063,20,0.05) |

*Table E2.* Hyperparameter settings across image restoration tasks. We report $(\mathrm{lr}, N_{\mathrm{optim}})$ for FlowGrad and for FlowDPS, where $N_{\mathrm{optim}}$ is the number of optimization steps, $(\lambda, \mathrm{lr}, \mathrm{max\text{-}iters})$ for FlowGrad, $(\lambda, \mathrm{lr}, \mathrm{max\text{-}iters}, \mathrm{weight\_decay})$ for OC-Flow, and $(\lambda, N_{\mathrm{ctrl}}, \mathrm{lr})$ for MPC-based methods, where $N_{\mathrm{ctrl}}$ denotes the number of control optimisation steps used to solve the optimisation problem in line 5 of Algorithms 1, 2 and 3.

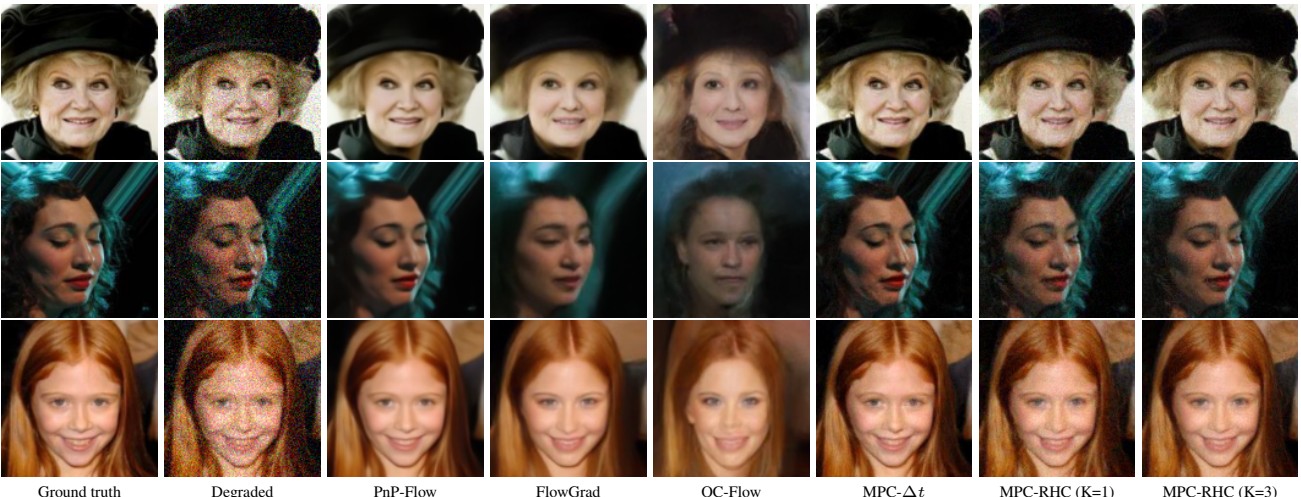

| Ground truth | Degraded | PnP-Flow | FlowGrad | OC-Flow | MPC-$\Delta t$ | MPC-RHC (K=1) | MPC-RHC (K=3) |
|---|---|---|---|---|---|---|---|

*Figure E1.* Qualitative results on the CelebA dataset for the denoising task with noise level $\sigma = 0.2$.

measured pixel-wise consistency between the decoded image and the target luminance image.

For style transfer, we varied $\rho$ from $4.0$ to $1024.0$ and used a fixed learning rate of $0.5$. For recoloring, we varied $\rho$ from $1 \times 10^{-8}$ to $5 \times 10^{-4}$ and used a learning rate of $0.2$.

The images in Figure 5 were generated with different prompts and reference style images for our method $\rho = 8.0$ for our method and learning rate $= 0.025$ for FlowChef.

**Model origin**    We used the 4-bit quantised model Flux.2 [dev] [3] obtained from the Diffusers Python package (von Platen et al., 2022). We make use of the Flux.2 [dev] model under the terms of the FLUX [dev] Non-Commercial License v2.0.

**Memory usage**    Inference-time conditioning can be run within tight GPU memory budgets by keeping heavyweight components off-GPU when they aren't needed. The components of interest are: text encoder (14.7 GB), latent space decoder (0.17 GB), Flux transformer (17.3 GB, quantised to 4 bits). In our setup, we move the text encoder to GPU to compute prompt embeddings, and then back to CPU. The Flux transformer and VAE are then moved to GPU for the denoising and control steps. This staged execution, combined with prompt-embedding caching and aggressive cache clearing between runs, keeps peak VRAM bounded below 24 GB even when using inference-time rewards.

**Dataset construction**    For style transfer, we used five prompts ("a cat", "a landscape", "a portrait", "a dog" and "two lizards wearing fedoras") paired with each of nine style images to construct a set of 45 style transfer tasks that we evaluated methods on. Images were generated at $448 \times 448$ resolution.

---
[3] https://huggingface.co/diffusers/FLUX.2-dev-bnb-4bit

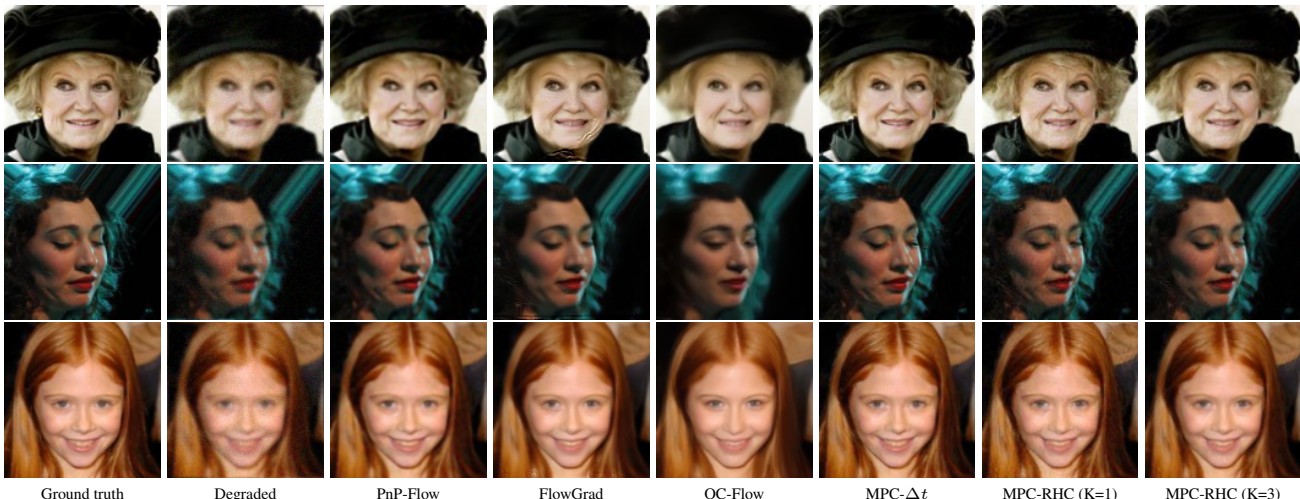

| Ground truth | Degraded | PnP-Flow | FlowGrad | OC-Flow | MPC-$\Delta t$ | MPC-RHC (K=1) | MPC-RHC (K=3) |

*Figure E2.* Qualitative results on the CelebA dataset for the image deblurring task with noise level $\sigma = 0.05$ and std of blurring Gaussian kernel set to $\sigma_b = 1.0$.

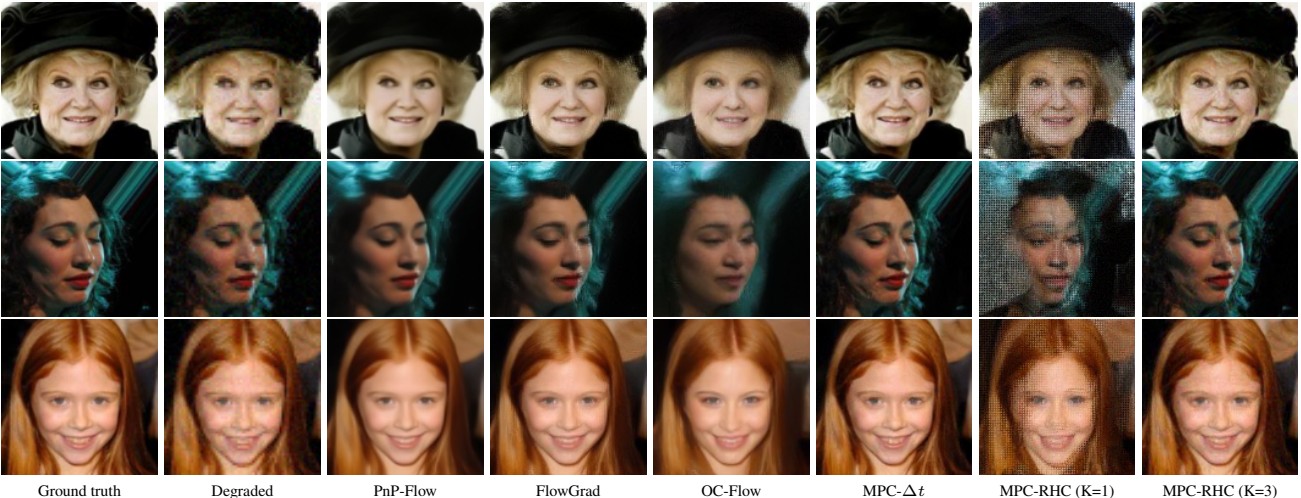

| Ground truth | Degraded | PnP-Flow | FlowGrad | OC-Flow | MPC-$\Delta t$ | MPC-RHC (K=1) | MPC-RHC (K=3) |

*Figure E3.* Qualitative results on the CelebA dataset for the image super-resolution task with noise level $\sigma = 0.05$ and $\times 2$ upscaling.

For image colouration, we used ten images sourced from Openverse [4], which were cropped and resized to $256 \times 256$ resolution.

In Figure F2, "lisa-the-dog" by roman.schurte is marked with CC0 1.0; "High School Ring" by slgckgc is licensed under CC BY 2.0; "Piano" by Sean MacEntee is licensed under CC BY 2.0; "shoes" by kats_stock_photos is licensed under CC BY 2.0; "The Iconic Image" by Andy Morffew is licensed under CC BY 2.0. -

## G. Image Restoration with Stable Diffusion 3.0

To demonstrate the flexibility of our framework across latent flow models, we qualitatively compare MPC-Flow with $K = 1$ against FlowDPS (Kim et al., 2025) and FlowChef (Patel et al., 2025), using Stable Diffusion 3.0 (SD3.0) (Esser et al., 2024). We evaluate the methods on three samples drawn from the DIV2K train set, the AFHQ train set, and the FFHQ train set, respectively, using the same examples provided in the official FlowDPS repository.[5] Following FlowDPS, we consider three inverse problems: (i) super-resolution (SR) from average pooling with a scale factor of 12, (ii) SR from

---

[4]https://openverse.org/
[5]https://github.com/FlowDPS-Inverse/FlowDPS

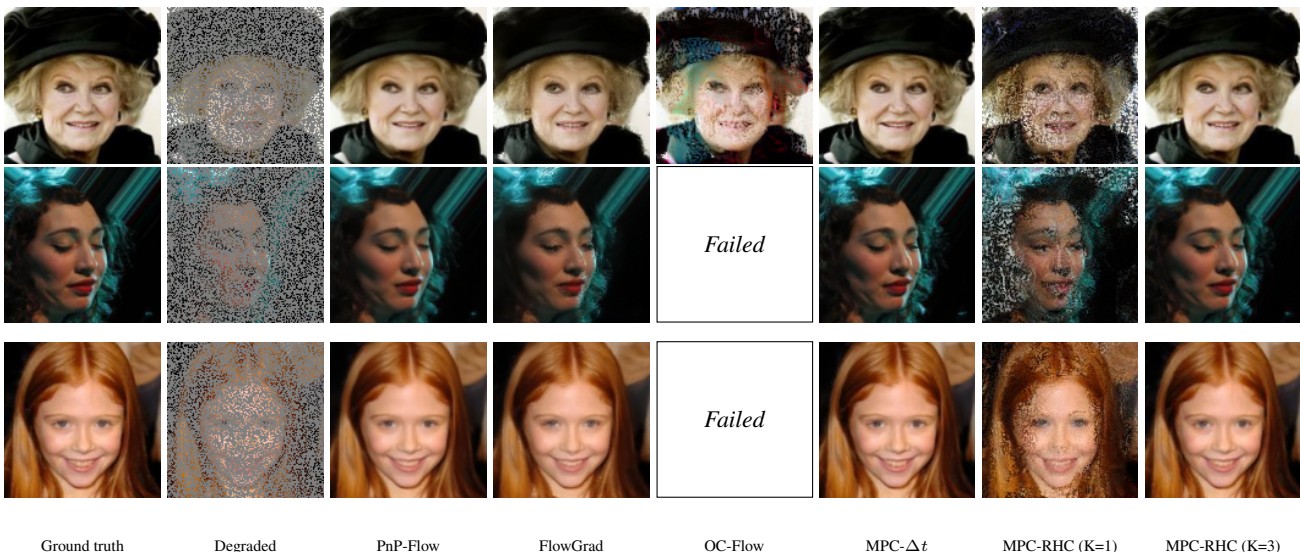

Ground truth | Degraded | PnP-Flow | FlowGrad | OC-Flow | MPC-$\Delta t$ | MPC-RHC (K=1) | MPC-RHC (K=3)

*Figure E4.* Qualitative results on the CelebA dataset for the random image inpainting task with noise level $\sigma = 0.05$ and 70% random masking. OC-Flow fails to produce meaningful restorations and is therefore indicated as failed.

---

**Algorithm 3** Single-Step RHC

---

**input** Pretrained flow model $v_\theta(\boldsymbol{x}, t)$, terminal loss $\Phi(\boldsymbol{x})$, initial state $\boldsymbol{x}_0$, control step size $\delta$

1: $t' \leftarrow 0, \quad \boldsymbol{x} \leftarrow \boldsymbol{x}_0$
2: **while** $t' < 1$ **do**
3:     Initialise control $\boldsymbol{u}$ (e.g., warm-start)
4:     **solve:**
5:        $\boldsymbol{u}^* = \arg\min_{\boldsymbol{u}} \left[ (1 - t') \|\boldsymbol{u}\|^2 + \Phi(\boldsymbol{x}') \right]$
6:        with: $\boldsymbol{x}' = \boldsymbol{x} + (1 - t')\big(v_\theta(\boldsymbol{x}, t') + \boldsymbol{u}\big)$
7:     Apply control $\boldsymbol{u}_0^*$ on interval $[t', t' + \delta]$
8:     Update state: $\boldsymbol{x} \leftarrow \boldsymbol{x} + \delta(v_\theta(\boldsymbol{x}, t') + \boldsymbol{u}^*)$
9:     Update time: $t' \leftarrow t' + \delta$
10: **end while**

---

bicubic interpolation with a scale factor of 12, and (iii) Gaussian deblurring with a kernel size of 61 and standard deviation 3.0. The resulting qualitative comparisons are shown in Figure F3.

## H. Interpretation as a Regulariser for Inverse Problems

Let us consider inverse problems with additive Gaussian noise, i.e.,

$$\boldsymbol{y} = \mathcal{A}(\boldsymbol{x}) + \boldsymbol{\epsilon}.$$

In this case, we can choose the terminal loss function as $\Phi(\boldsymbol{x}) = \|\mathcal{A}(\boldsymbol{x}) - \boldsymbol{y}\|^2$, measuring the consistency of the reconstruction $\boldsymbol{x}$ to the observations $\boldsymbol{y}$. Let us consider the global optimal problem (5), which we re-state here as

$$\min_{\boldsymbol{u}} \left\{ \mathcal{J}_\rho(\boldsymbol{u}) = \|\mathcal{A}((\boldsymbol{x}(1)) - \boldsymbol{y}\|^2 + \rho \int_0^1 \|\boldsymbol{u}(\tau)\|_2^2 d\tau \right\}$$

$$\text{s.t.} \quad d\boldsymbol{x}(t)/dt = v_\theta(\boldsymbol{x}(t), t) + \boldsymbol{u}(t),$$
$$t \in [0, 1], \boldsymbol{x}(0) = \boldsymbol{x}_0,$$

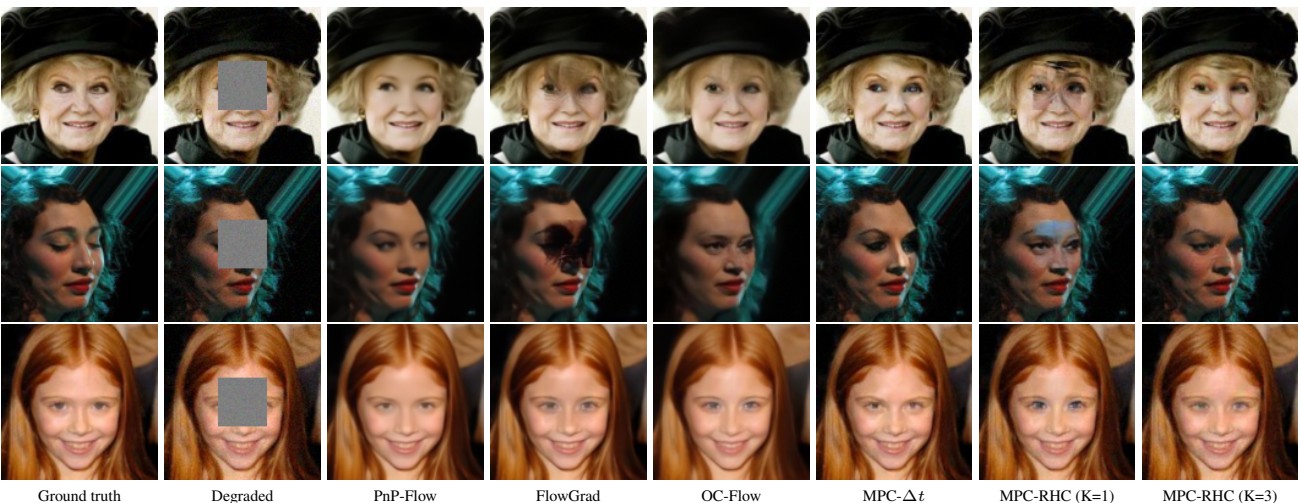

| Ground truth | Degraded | PnP-Flow | FlowGrad | OC-Flow | MPC-$\Delta t$ | MPC-RHC (K=1) | MPC-RHC (K=3) |

*Figure E5.* Qualitative results on the CelebA dataset for the box inpainting task with noise level $\sigma = 0.05$, and a central missing region of size $40 \times 40$ pixels.

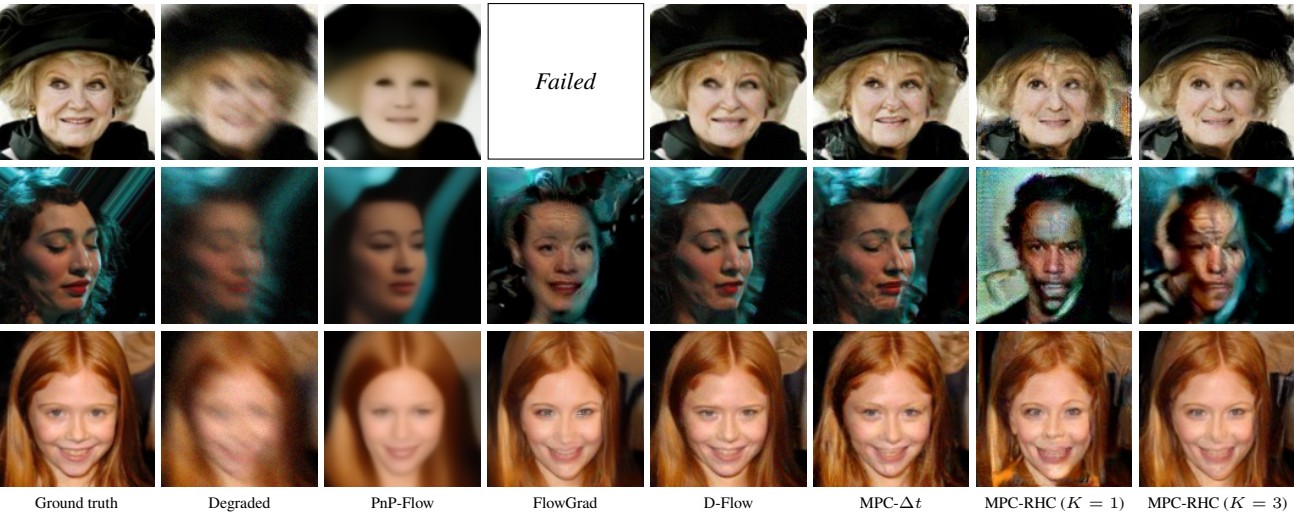

| Ground truth | Degraded | PnP-Flow | FlowGrad | D-Flow | MPC-$\Delta t$ | MPC-RHC ($K = 1$) | MPC-RHC ($K = 3$) |

*Figure E6.* Qualitative results on the CelebA dataset for the nonlinear blur task with noise level $\sigma = 0.05$.

with $\rho = 1/\lambda$. We can now interpret this as a classical regularisation functional in the variational regularisation framework (Engl et al., 1996). For this, we consider the objective function

$$\arg\min_{\boldsymbol{x}_1} \|\mathcal{A}(\boldsymbol{x}_1) - \boldsymbol{y}\|^2 + \rho R(\boldsymbol{x}_1), \tag{19}$$

with $R$ defined as the solution to a constrained optimal control problem

$$R(\boldsymbol{x}_1) = \inf_{\boldsymbol{u}} \left\{ \int_0^1 \|\boldsymbol{u}(\tau)\|^2 d\tau \,\middle|\, \frac{d\boldsymbol{x}_t}{dt} = v_\theta(\boldsymbol{x}_t, t) + \boldsymbol{u}(t), \quad \boldsymbol{x}(0) = \boldsymbol{x}_0, \quad \boldsymbol{x}(1) = \boldsymbol{x}_1 \right\}. \tag{20}$$

This can be interpreted as a type of dynamic prior, which penalises $\boldsymbol{x}_1$ if it is not *easily reachable*, i.e., there does not exist a trajectory with a small control, from $\boldsymbol{x}_0$. In this way, the flow model defines as distance

$$d_{v_\theta}(\boldsymbol{x}_1, \boldsymbol{x}_0) = \inf_{\boldsymbol{u}} \left\{ \int_0^1 \|\boldsymbol{u}(\tau)\|_2^2 d\tau \,\middle|\, \frac{d\boldsymbol{x}_t}{dt} = v_\theta(\boldsymbol{x}_t, t) + u(t), \boldsymbol{x}(0) = \boldsymbol{x}_0, \boldsymbol{x}(1) = \boldsymbol{x}_1 \right\}, \tag{21}$$

which explicitly highlights the influence of the initial value $\boldsymbol{x}_0$. We get an interesting special case if we set the flow model to zero, i.e., $v_\theta = 0$. In this case the dynamics reduce to $\frac{d\boldsymbol{x}_t}{dt} = \boldsymbol{u}(t)$ and the optimal solution is simply a straight line from $\boldsymbol{x}_0$ to $\boldsymbol{x}_1$ and the distance reduces to $d_0(\boldsymbol{x}_1, \boldsymbol{x}_0) = \|\boldsymbol{x}_1 - \boldsymbol{x}_0\|^2$ and we obtain a typical Tikhonov regularisation functional.

GT target

LR measurement x8

Unconditional
Proj PSNR=13.56 dB
1-LPIPS=0.426

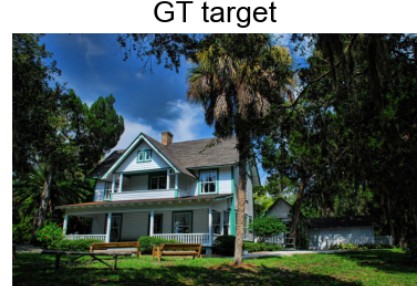
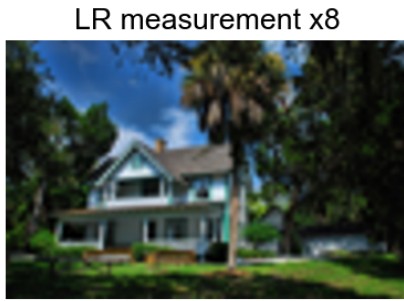
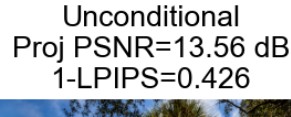
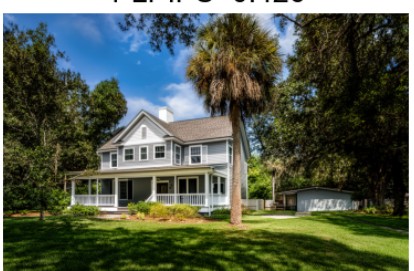

MPC-Flow best projected consistency
Proj PSNR=30.46 dB

MPC-Flow best perceptual
1-LPIPS=0.530

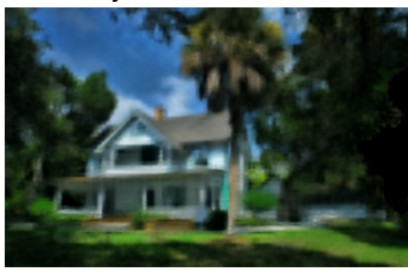
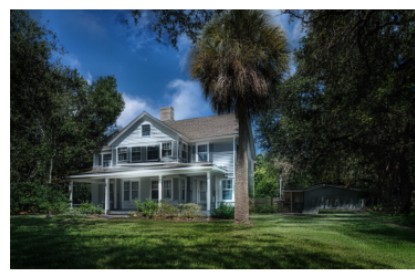

FlowChef best projected consistency
Proj PSNR=28.03 dB

FlowChef best perceptual
1-LPIPS=0.485

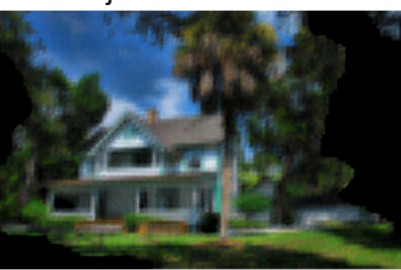
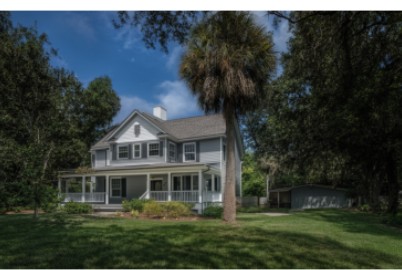

*Figure F1.* Example images for 8x super-resolution task with FLUX.2. Proj PSNR refers to the PSNR of the low-resolution image as measured in the high-resolution space (by upsampling with the adjoint of the downsampling operator).

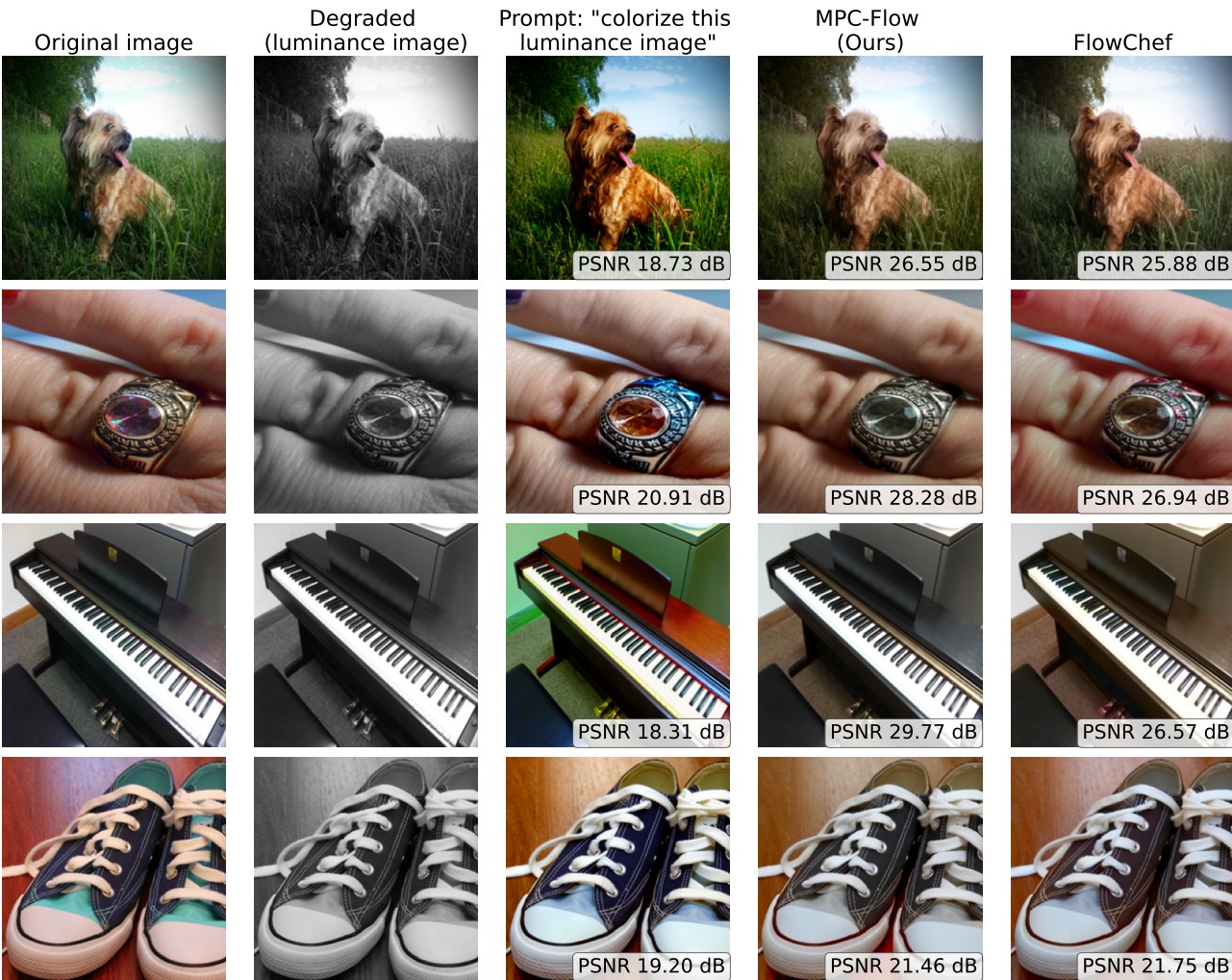

*Figure F2.* Further examples of image recolouration with Flux.2 and MPC-Flow.

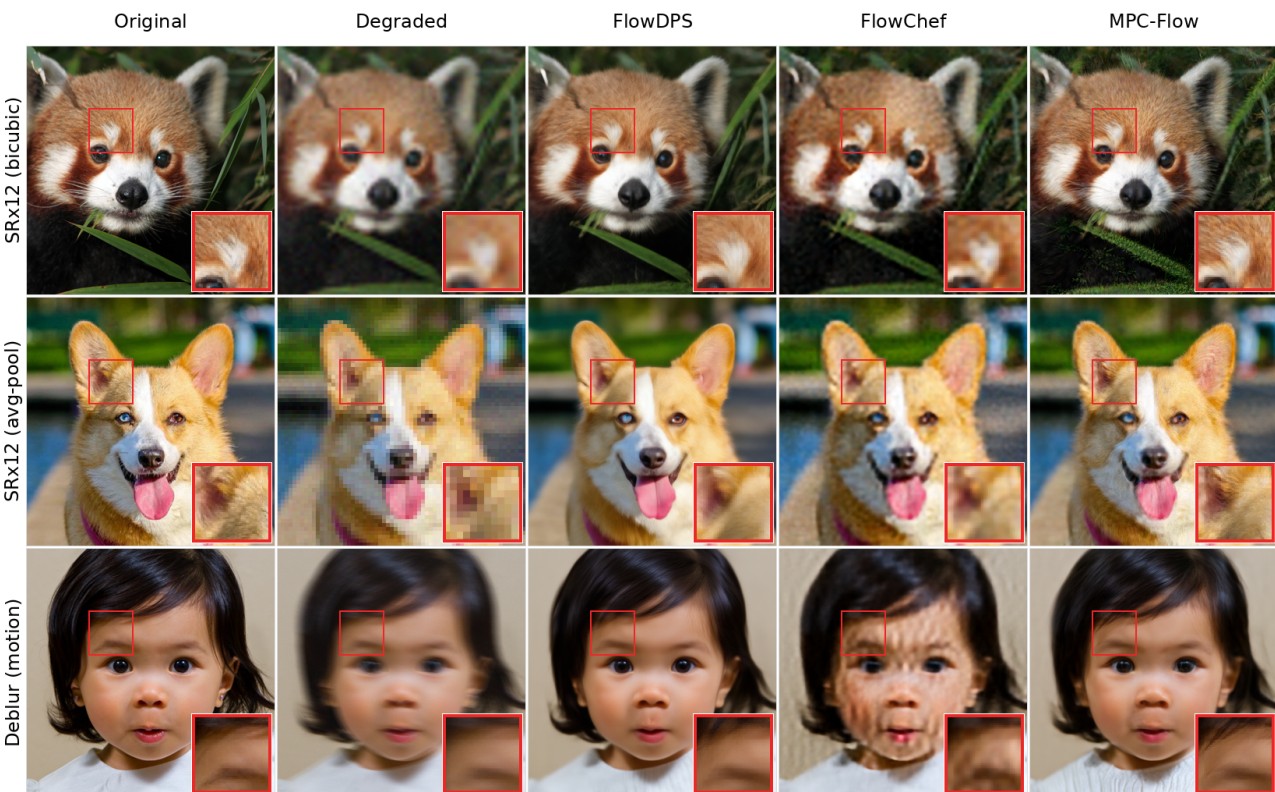

*Figure F3.* Qualitative comparison of MPC-Flow with $K = 1$ against FlowDPS (Kim et al., 2025) and FlowChef (Patel et al., 2025). The first, second, and third rows correspond to samples from the DIV2K train set, AFHQ train set, and FFHQ train set, respectively. Following the FlowDPS evaluation setup, the three tasks are: super-resolution from average pooling with scale factor 12, super-resolution from bicubic interpolation with scale factor 12, and Gaussian deblurring.

