# OpenReview forum: "Solving Inverse Problems with Flow-based Models via Model Predictive Control"
_ICML.cc/2026/Conference — ICML 2026 regular_

### Official Review · Reviewer_zEac · 2026-03-10

**Soundness:** 4
**Presentation:** 4
**Significance:** 3
**Originality:** 2
**Overall Recommendation:** 5
**Confidence:** 4

**Summary:**

This paper introduces MPC-Flow, a guidance mechanism for flow-based generative models which is grounded in optimal control. The authors claim that this paper is the first conditional generation method rooted in optimal control. Two methods of casting flow matching into optimal control are proposed: planning until the end of the trajectory or one step planning. One step planning removes the need to differentiate through the flow model which yields better computational performance, allowing for scalability to large models.

The authors demonstrate their proposed framework on several tasks including style transfer and image super-resolution. Their results show that MPC-Flow is able to solve these tasks.

**Compliance With Llm Reviewing Policy:**

Affirmed.

**Key Questions For Authors:**

- For the \Delta t control you mention that the choice in cost function is equal to the value function itself. You approximate the value function as the terminal cost evaluated at the straight line distance between the current state and the terminal state. However, usually in value functions the controls in the future are incorporated. For example: the cost of the control from "now" until the terminal time under the current policy. Is this the source of the approximation error in Equation 12?
- Is the following paper relevant in your work? It seems the authors also cast flow matching as an optimal control problem here: https://arxiv.org/pdf/2510.02315
- Do the authors have any insight into why larger K does not seem to improve performance? In optimal control planning is performed over action sequences. Planning for larger horizons can usually increase control performance.

**Limitations:**

Yes

**Strengths And Weaknesses:**

Strengths:
- The paper is clearly high quality and presented well.
- The authors compare against other flow-based optimal control models and provide numerical results demonstrating the benefit of their model.
- The authors demonstrate that their method can scale to large generative models with billions of parameters.

Weaknesses:
- The title and introduction frames the paper as solving inverse problems. However, the method and experiment section the paper seems to focus more on integrating MPC into flow matching in order to guide the output of the generative model. This appears to be a slight contradiction.
- For ease of reading Table 2 would benefit from arrows indicating if larger or smaller numbers are better.
- Limited insight into planning over longer horizons. I understand that with large flow based models long horizon planning is not tractable. However, perhaps the inclusion of an experiment on a low dimensional example could improve the insight of the paper.
- While the technical formulation is sound, the degree of novelty relative to prior optimal-control formulations of generative models appears moderate.

Textual bugs:
- Missing a space between the period and the T "through the flow model vθ.This dramatically".
- Potential typo? "UNet with approx. 8Mio parameter, which allows us to directly implement the global optimal"

---

> ### Author Rebuttal · Authors · 2026-03-30
>
> We thank the reviewer for the insightful feedback and agree that our approach exposes a class of algorithms with a built-in accuracy–compute trade-off.
>
> ---
> **MPC-Flow for solving inverse problems:** We agree that the goal of MPC-Flow is to guide the output of a generative model. However, this guidance is precisely the mechanism used to solve inverse problems. In our formulation, MPC-Flow provides a principled way to steer flow-based models towards solutions that satisfy the constraints imposed by the inverse problem (i.e., consistency to measurements). This is the primary focus of our experiments on computed tomography, image restoration, and image colouration (using FLUX.2), all of which are naturally formulated as inverse problems. The style transfer experiment does not constitute a typical inverse problem. Its purpose is instead to demonstrate that the proposed framework extends beyond inverse problems to more general guidance settings, where the task can be formulated as an optimal control problem. We will revise the experimental section to clarify this connection.
>
>
> On a minor note, **we do not claim to be the first to use optimal control for conditional generation**. Prior works, e.g., Liu et al. (2023) [1] and Wang et al. (2024) [2], are discussed in the background. To the best of our knowledge, however, our approach, particularly MPC-RHC ($K=1$), is the first optimal control-based method to scale to large models such as FLUX.2 and Stable Diffusion 3.0.
>
> ----
>
> We now address the reviewer’s questions:
>
> **Q1.** As we state in Theorem 3.2 the optimal choice for the cost function for MPC-$\Delta t$ would be to use the value function. As the reviewer points out, the value function depends on the future controls and trajectory and is thus not tractable. This is why we approximate the value function using a single Euler-step approximation (see Eq. 12 in our manuscript). For the toy problem in Appendix C, we can compute the value function by solving the corresponding HJB equation. Please see our response to reviewer AEUf (Q2), where we explictly show the error of our single-step approximation of the value function. Note, that this error decreases for $t \to 1$.
>
> **Q2. Relation to Bill et al. (2025).** The work [3] is related in that it also cast guided generation of flow models as an (stochastic) optimal control problem. However, the objective/task and resulting control strategies differ from ours. [3] focus on multi-subject disentanglement for text-to-image models, which they formulate as a stochastic optimal control problem. They propose both a test-time method and a fine-tuning approach. Since our MPC variants are test-time methods, the most relevant comparison is with their test-time formulation. Their method approximates the adjoint state and, in the deterministic limit (their Eq. 15–17), considers an optimal control objective of the form $$ \min_u \int_0^1 \frac{1}{2} \| u(X_t, t) \|^2 + f(X_t, t)dt $$ where $f(X_t,t)$ is a running cost applied throughout the trajectory. A main contribution of [3] is to define a useful running cost for their application (termed FOCUS).
> In our inverse problem setting, where the goal is for the terminal state to match the measurements $y$ (i.e., $\Phi(X_1) = | A X_1 - y |^2$), we cannot apply the approximation of [3], as there is no natural choice of a running cost $f$.
>
> **Q3. Performance under changing $K$.** We refer the reviewer to the computed tomography experiments, in particular Figure 3. There, we compare the performance of MPC-RHC for varying $K$ (i.e., the number of discretisation steps). We observe that a larger $K$ (i.e., a finer discretisation of the remaining horizon) leads to improved performance in the computed tomography example.
>
> Furthermore, in the image restoration experiments reported in Table 2, we observe that $K=3$ yields better reconstruction quality than $K=1$, again with higher memory usage and computational cost. As discussed in the manuscript, the choice of $K$ reflects a trade-off between reconstruction quality, computational speed, and peak VRAM usage: increasing $K$ improves quality but increases resource requirements.
>
> ---
> Further points:
> - We fixed the typos in *Textual bugs*. Thank you for catching these!
> - We added arrows to Table 2.
>
> ---
> [1] Liu *et al.* "Flowgrad: Controlling the output of generative odes with gradients" (https://ieeexplore.ieee.org/document/10204490)
>
> [2] Wang *et al.* "Training Free Guided Flow Matching with Optimal Control"
> (https://arxiv.org/abs/2410.18070)
>
> [3] Bill *et al.* "Optimal Control Meets Flow Matching: A Principled Route to Multi-Subject Fidelity" (https://arxiv.org/pdf/2510.02315)

---

> > ### Author Rebuttal · Reviewer_zEac · 2026-04-04
> >
> > The authors have fully addressed my concerns. While this strengthens my assessment of the paper, I cannot raise my score further because, in my view, the next score range is reserved for papers with truly field-changing impact. Although this paper has clear strengths, I do not believe it reaches that level.

---

### Official Review · Reviewer_QPAM · 2026-03-12

**Soundness:** 3
**Presentation:** 3
**Significance:** 3
**Originality:** 3
**Overall Recommendation:** 4
**Confidence:** 4

**Summary:**

This paper views the inverse problem as a control problem under flow-based generative model’s setting. The previous work considers it as an optimal control problem, while effective, it is computationally intense. This paper addresses this by breaking the problem into a sequence of smaller, manageable "short-horizon" sub-problems.

**Compliance With Llm Reviewing Policy:**

Affirmed.

**Key Questions For Authors:**

Please refer to Strengths And Weaknesses

**Limitations:**

Yes

**Strengths And Weaknesses:**

This paper reframes conditional generation in flow models as a sequence of short-horizon optimal control sub-problems, rather than relying on heuristic guidance strategies. The method avoids the prohibitive memory costs of differentiating through the entire ODE sampling dynamics or relying on costly adjoint methods. While highly scalable and theoretically sound, I have some concerns:

1. The optimal control trajectory is highly dependent on the initial value $x_0$. Do authors have tried any initialization strategy (e.g. DFlow’s initialization) to obtain better performance, considering that the Single-Step RHC performs poorly on linear tasks.

2. To achieve high performance on the tasks where the $K=1$ variant fails, the framework must use $K>1$ discrete steps. There is still a non-trivial computational overhead.

3. The proposed method seems to rely heavily on the linear flow / rectified flow. How does the curvature of the flow trajectory affect the performance (e.g. rectified flow and plain flow model)?

4. Since the proposed method shares some similarities with FlowChef, can the authors show the comparison with FlowChef in image restoration tasks using Flux or Stable Diffusion?

---

> ### Author Rebuttal · Authors · 2026-03-30
>
> Thank you for the positive review and helpful questions. We will now address your questions and concerns:
>
> **Q1. Dependence on initial value:** We agree that the optimal control trajectory can be sensitive to initialisation, and that more informed initialisation strategies (such as the one in D-Flow) could improve performance, particularly for the single-step MPC-RHC ($K=1$) setting. In the current manuscript, we intentionally use random initialisation for all MPC-Flow experiments to avoid introducing additional hyperparameters, and to demonstrate that the proposed method can perform even under this minimal setup.
> The scope of this work is to introduce a class of algorithms in a principled way, rather than to achieve the best possible reconstruction. We therefore chose not to use D-flow initialization, as it is more heuristic and may introduce additional confounding factors. We will clarify this point in the revised manuscript and highlight initialisation as an important direction for improving performance and for future work.
>
> For the D-Flow results reported in Table 2, we follow the standard D-Flow initialisation procedure.
>
> **Q2. Computational overhead:** We agree that using multiple MPC steps ($K>1$) increases computational cost relative to some lightweight approaches. However, our methods remain substantially cheaper than those that estimate the global optimal control directly. This overhead is inherent to the framework, which exposes a controllable trade-off between compute and solution quality: $K=1$ eliminates backpropagation and is highly efficient, while larger $K$ improves accuracy at increased cost, still remaining significantly cheaper than full-horizon optimal control methods.
>
> **Q3. Dependence on flow geometry:** We agree that the effectiveness of one-step approximations depends on the curvature of the flow trajectory: for straighter trajectories (e.g., rectified flows), local approximations are more accurate. However, this is not a limitation of the framework itself, but of the low-$K$ regime. Increasing $K$ improves accuracy regardless of flow curvature by refining the approximation to the globally optimal solution. Empirically, we observe consistent behaviour across both standard flow-matching models (linear flow for CT, and OT-flow for CelebA) and rectified flows (FLUX.2). We will clarify this relationship between flow geometry and the $K$-dependent approximation in the manuscript.
>
> **Q4. Comparison to FlowChef:** We agree this is an important comparison. We have:
> - Added FlowChef along with FlowDPS [1] to our image restoration experiments on CelebA (see https://bashify.io/i/6rCUMg). These methods were originally designed for latent diffusion reconstruction tasks and lack explicit regularization strategies, resulting in suboptimal performance. Further, FlowChef was developed for rectified flows whereas our CelebA model is an OT-flow model. We also add the LPIPS for all methods.
> - Added an additional super-resolution experiment on FLUX.2 comparing FlowChef with our proposed method (see https://bashify.io/i/6ry4f0).
> - Added further qualitative inverse problem experiments (super-resolution and motion deblurring) comparing our method against FlowDPS and FlowChef using Stable Diffusion 3.0 (see https://bashify.io/i/XazdAh) on samples taken from the DIV2K validation set.
>
> These results indicate that our method achieves improved trade-offs between perceptual quality and measurement fidelity, due to the path regularization introduced by MPC-Flow.
>
> Conceptually, while the resulting algorithms share similarities, our approach is derived from an MPC formulation of the underlying optimal control problem. This provides a principled objective and a systematic framework for constructing approximations, including an explicit mechanism (via $K$) to trade off accuracy and compute. In contrast, comparable methods rely on fixed heuristic designs without such a unifying control perspective. We will clarify both the empirical comparison and this distinction in the revision.
>
> [1] J. Kim *et al.* "Flow-Driven Posterior Sampling for Inverse Problems"
> (https://arxiv.org/abs/2503.08136)

---

> > ### Author Rebuttal · Reviewer_QPAM · 2026-04-01
> >
> > The authors have provided a strong and thorough rebuttal that satisfactorily addresses my concerns.
> >
> > However, there is one critical issue I need to point out: the examples shown in the [external link](https://bashify.io/i/XazdAh) are **not** from the DIV2K validation set as claimed in the rebuttal. They are actually from the DIV2K training set, FFHQ, and AFHQ. Furthermore, these are the exact same examples found in the [FlowDPS repository](https://github.com/FlowDPS-Inverse/FlowDPS/tree/main/samples).
> >
> > I strongly urge the authors to be much more *rigorous* and *honest* about their experimental setups and the source of their evaluation data.

---

> > > ### Author Response · Authors · 2026-04-01
> > >
> > > Thank you for pointing out this oversight, and we sincerely apologise for the confusion.
> > >
> > > You are absolutely correct: the examples shown in the external link were not from the DIV2K validation set as stated in the rebuttal.
> > > To be more precise: the first row is from DIV2K (train), the second row is from AFHQ (train) and the last row is from FFHQ (train).
> > >
> > > Further, we referred the image in the first row as *DIV2K validation* due to the caption of Figure 3 in the FlowDPS paper (https://arxiv.org/pdf/2503.08136).
> > >
> > > To clarify, for this additional experiment we re-implemented a minimal version of MPC-Flow (RHC $K=1$) within the FlowDPS repository to have a direct comparison on their provided examples. We are happy to share this implementation (as an additional method in the *sd3_sampler.py* file of FlowDPS): https://anonymous.4open.science/r/MPCFlowSD30BC66/README.md. Our intention here was not to provide additional rigorous experimental comparison, but rather to provide a *qualitative* investigation on a very limited number of samples to demonstrate that MPC-Flow can also be run using StableDiffusion 3.0 as a response to the question of Reviewer bvcg (Q1).
> > >
> > > Importantly, all quantitative results in the paper are computed on the specified datasets and splits (e.g., CelebA test set), and are unaffected.

---

### Official Review · Reviewer_bvcg · 2026-03-12

**Soundness:** 2
**Presentation:** 3
**Significance:** 2
**Originality:** 2
**Overall Recommendation:** 4
**Confidence:** 4

**Summary:**

The optimal control problem is for obtaining a control u that minimizes integration of energy of u across the entire ODE trajectory and terminal cost. As the problem requires backpropagation through roll-out predictions, model predictive control (MPC) has been proposed in the literature of control theory.

The paper leverages MPC to flow-model-based inverse problem solver, which is recently formulated as the optimal control problem. The main idea of MPC is to divide the optimal control problem into smaller problems defined for each trajectory segment.

When computing the control at time t, if the cost function is integrated until the endpoint of flow ODE, the paper defines it as receding horizon control (RHC). If the cost function is integrated until the subsequent timepoint, the paper defines it as delta-t control.

In the case of RHC, the paper discretizes the time axis from t to the endpoint by K-steps. Then, compute the cost function by solving remaining ODE during K steps, compute the terminal cost using the solution, and finally update the control variable, which is applied for a small timestep. When K >1, this approach requires backpropagation through K roll-out processes. However, when K=1, it does not require backpropagation as the closed form solution for control is obtained.

In the case of delta-t control, the terminal cost is evaluated at Tweedie estimate, which is an approximation for the optimal value function. This also does not require backpropagation as the closed form solution for control is obtained.

For multiple inverse problems designed on CelebA dataset, the proposed method outperforms optimal-control driven methods. Also, the paper has demonstrated extended applications such as image style transfer and colorization.

**Compliance With Llm Reviewing Policy:**

Affirmed.

**Final Justification:**

The authors' responses have resolved major concerns so the reviewer increases score from 3 to 4.

**Key Questions For Authors:**

- Have authors applied the proposed method to Stable Diffusion 3.0 or Flux.1 (dev) when solving inverse problems, as related works have done? In other words, is the proposed method generalizable to any flow models as the title says?

- Have authors analyzed the error induced by MPC-RHC (K=1) or delta-t control? The reviewer finds that the property in lines 196-200 (right column) looks great, but the proposed method loses it by emphasizing K=1 case.

- Have authors compared the proposed MPC-based solver with other approaches including diffusion posterior sampling (FlowDPS), variational inference (FLAIR), or stochastic monte carlo (DAPS)? Does the optimal control have benefits over those approaches? Note that none-of those approaches do backpropagation through the flow model.

- What about applying the delta-t control approach to applications in section 4.3? For inverse problems, it is better than MPC-RHC (K=1) as shown in Table 2, but is not included in Table 3.

**Limitations:**

yes

**Strengths And Weaknesses:**

**Strength**

- The paper is clearly written and easy to follow.

- The paper is self-contained and well describes background knowledge on MPC.

- The paper proposes a novel and useful viewpoint for inverse problem solvers.

- The proposed method, especially MPC-RHC (K=1), is simple and efficient, yet effective.

**Weakness**

- Concern on the accuracy of one-step cost evaluations: While MPC-RHC (K=1) and delta-t control are highly efficient as they only requires one-step feed-forward of flow model to obtain a good control variable, it inherently has large approximation error, especially at high-SNR regions. When the flow model has relatively straight trajectory (e.g. FLUX), this approximation can be practical, but the reviewer has concern whether those approaches can be used in flow models in general.

- Regarding the first weakness, the reviewer has concerns about limited experiments. 1) Current inverse problems are only tested with the OT-flow model, which may have a straight trajectory, while related works in this area have explored stable diffusions or FLUX. Note that the proposed method could be readily implemented on those models without any changes except model. 2) Experiments are done only with human face data (CelebA). It would be stronger to add a more diverse dataset like ImageNet or DIV2K. 3) The paper does not include other flow-model-based inverse problem solvers. To demonstrate the advantage of the optimal control theory in inverse problem, comparison with other approaches based on diffusion posterior sampling or stochastic monte carlo simulation should be compared together.

- Except for the MPC-RHC (K=1) case, the MPC-driven approach needs a larger number of function evaluations due to repeated roll-out at each timepoint, as shown in table 2.

- Clarification for delta-t control: The reviewer finds a gap between equation (11) and algorithm 2. The reviewer especially couldn’t find discussion about how small the delta_t is, and how the integration from t to t+delat_t in equation (11) disappears.

- Lines 325-327 (left column) needs clarification.

---

> ### Author Rebuttal · Authors · 2026-03-30
>
> Thank you for this detailed review. We address concerns on one-step approximations, experimental coverage and comparisons, and clarify MPC-$\Delta t$ and implementation details.
>
> **W1. Accuracy of one-step evaluation:** We agree that MPC-RHC ($K=1$) and MPC-$\Delta t$ introduce approximation error, as they locally approximate a global optimal control problem. However, MPC-RHC ($K=1$) is one point on a broader algorithmic spectrum: $K=1$ corresponds to the lowest-compute regime, while increasing $K$ systematically refines the solution.
>
> Importantly, *one-step approximations are not unique to our approach*. Many existing flow-based inverse-problem solvers also rely on local or single-step updates to remain computationally tractable. Our contribution is to extend optimal-control-based methods to this regime, while making the approximation explicit and controllable via the MPC framework, exposing a clear accuracy–compute trade-off. Explicit control of the approximation level is a key distinction from prior methods.
>
> Empirically, $K=1$ remains effective across models (CT, CelebA, FLUX.2), and Appendix C shows that increasing $K$ reduces the gap to the optimal policy. We will clarify this trade-off in the revision (see also response to Reviewer AEUf, "Theory-Practice Gap").
>
> **W2. Experimental coverage:** We agree that broader coverage would further strengthen validation, and we have extended our experiments accordingly.
>
> - _Flow Model Diversity:_ In addition to the inverse problem tasks in CT (linear flow) and CelebA (OT-flow), we now include super-resolution results for the FLUX.2 model (see https://bashify.io/i/6ry4f0 and the response to Reviewer QPAM) as well as super-resolution and motion deblurring tasks for StableDiffusion 3.0 (see https://bashify.io/i/XazdAh) on samples taken from the DIV2K validation set, demonstrating applicability across different flow model classes.
> - _Method Comparisons:_ we compare against optimal-control methods (OC-Flow, FlowGrad) and flow-based solvers (OT-ODE, D-Flow, Flow-Priors, PnP-Flow), and now additionally include FlowChef and FlowDPS. We also attempted to incorporate FLAIR, however, adapting its latent-space implementation to our pixel-space setting was unstable.
>
> **W3. Number of NFEs:** We agree that for $K>1$ the MPC approach requires more NFEs than some lightweight comparator methods due to repeated rollouts. However, compared to methods that optimise a global control over the full trajectory (e.g., OC-Flow, FlowGrad), MPC remains significantly more efficient. In particular, for CelebA even MPC with $K=3$ ($175$s per image) is faster than OC-Flow ($234$s per image) and Flow-Grad ($315$s per image). This reflects the core contribution: a controllable trade-off between compute ($K$) and approximation quality using optimal control.
>
> **W4. Clarify $\Delta t$ Control:** In Eq. (11), the cost is defined over $[t, t+\Delta t]$. In Algorithm 2, this is discretised with a single Euler step, so the integral becomes $\Delta t \|u\|^2$, yielding a one-step optimisation with a value-function surrogate for the remaining cost. Here $\Delta t = 1/N$. We will clarify this in the revision.
>
> **W5. Lines 325-327.** We agree this statement is unclear. The intended meaning is that FLUX.2 and related flow models do not enforce task-specific constraints (e.g., measurement consistency) during generation, and therefore require control or guidance. We will revise this passage.
>
> ---
>
> ### Key Questions
>
> **Q1.** The method applies to general flow-based models. In our current manuscript we already employ the method on linear flows (CT, Section 4.1), OT-Flow (CelebA, Section 4.2) and rectified flows (FLUX.2, Section 4.3). To further make this point we provide a quantitative experiment using the setup from from FlowDPS (12x super-resolution and motion deblurring) on SD3.0 ('stabilityai/stable-diffusion-3-medium-diffusers'), please see this link https://bashify.io/i/XazdAh.
>
> **Q2.** MPC-RHC ($K=1$) introduces a structured approximation (see W1). Appendix C shows that increasing $K$ reduces the gap to the optimal policy, quantifying the accuracy–compute trade-off. The theoretical properties correspond to the continuous limit as $K$ increases.
>
> **Q3.** We now include FlowDPS and FlowChef on CelebA. Our comparisons focus on optimal-control and flow-based solvers to isolate the MPC contribution. The key advantage is a principled control objective with an explicit mechanism ($K$) to trade off accuracy and compute. Please see https://bashify.io/i/6rCUMg for the new Table. Note that FlowChef is developed for rectified flow models, whereas CelebA uses an OT-flow model. We will also comment on this discrepancy in the revised manuscript.
>
> **Q4.** In Table 3 (FLUX.2), we use a quantised model, making backpropagation unstable. Accordingly, we restrict to methods that avoid backpropagation in this setting. Since MPC-$\Delta t$ requires gradients through the model, we didn't include it in Section 4.3.

---

> > ### Author Rebuttal · Reviewer_bvcg · 2026-04-04
> >
> > Thanks to the authors for their kind response. The reviewer appreciates that the authors conducted additional experiments on additional tasks and datasets.
> >
> > In the previous comment, the reviewer missed the quantitative result that the authors provided. The reviewer apologizes for this oversight.
> >
> > At this stage, the major concerns have been properly addressed. If possible, the reviewer would recommend adding DAPS, an SMC-based approach, to the revised version.
> >
> > To further strengthen the positioning of this paper, it would be important to clarify the benefits of using optimal control theory over alternative theoretical frameworks. If possible, the reviewer also suggests adding more discussion on this aspect, in addition to the empirical performance.

---

> > > ### Author Response · Authors · 2026-04-06
> > >
> > > Thank you for your follow-up and for acknowledging our rebuttal. We appreciate the time you have taken to review our additional experiments.
> > >
> > > Regarding your final suggestions, we will address them in the revised version:
> > >
> > > **SMC-based Methods:** We appreciate the suggestion to include DAPS. We will discuss DAPS in the background and related work sections to provide a more complete picture of the current landscape of using generative models for inverse problems.
> > >
> > > **Theoretical Positioning:** Thank you for raising this point. In the revised manuscript, we will expand the discussion to clarify the advantages of the Optimal Control (OC) framework over methods like Diffusion Posterior Sampling (DPS) [1]:
> > >
> > > (1) *Spending Compute for Accuracy:* Methods like DPS rely on the Tweedie estimate (a one-step estimate of the final denoised sample) to steer the sampling path. Currently, there is no approach in that framework to "spend" more compute to achieve a more accurate estimate. Here, the OC framework has distinct advantages: by choosing a better discretization (a larger K for MPC-RHC), we can trade off accuracy vs. compute. While $K=1$ offers a one-step lookahead similar to what is used in DPS-style methods, $K>1$ provides a more accurate (but more expensive) lookahead.
> > >
> > > (2)*Regularisation:* By incorporating the regulariser $\int \| u(t) \|^2 dt$, we can directly control how closely the solution follows the unconditional trajectory. This provides an approach to balance measurement consistency with the learned prior.
> > >
> > > (3) *Similarities to other frameworks*: We will address more directly how other algorithms (like DPS above) can be seen as specific interpretations of the OC frameworks by choosing specific discretizations, approximations to the value functions and parametrisations of the control $u$.
> > >
> > > Given these clarifications and your confirmation that our response has addressed your major concerns, we would be grateful if you would consider raising your score to reflect this revised assessment. Thanks again for your review and helpful feedback.
> > >
> > > [1] Chung et al. "Diffusion Posterior Sampling for General Noisy Inverse Problems"

---

### Official Review · Reviewer_AEUf · 2026-03-15

**Soundness:** 3
**Presentation:** 4
**Significance:** 3
**Originality:** 2
**Overall Recommendation:** 3
**Confidence:** 3

**Summary:**

This paper studies training-free conditioning of flow-based generative models for inverse problems through model predictive control.

A core issue investigated by the study is how to retain the principled optimal-control formulation of conditional flow sampling without paying the full memory/runtime cost of optimizing an entire controlled trajectory.

The paper introduces MPC-Flow, a framework that solves global control of generative trajectories by repeatedly optimizing short-horizon subproblems. It analyzes two regimes, receding-horizon control and $\delta t$-horizon control, and proposes an efficient $K=1$ variant that avoids backpropagation through the full trajectory. The paper also presents Bellman-style optimality statements, connects the objective to variational regularization, and demonstrates results on toy/CT tasks, CelebA restoration, and FLUX.2 editing on 24GB GPUs.

**Compliance With Llm Reviewing Policy:**

Affirmed.

**Key Questions For Authors:**

1. Implementation of $K=1$. Please clarify exactly how MPC-RHC ($K=1$) is implemented. Is it a one-shot optimization over the remaining horizon followed by a single update to $t=1$ (as Algorithm 3 seems to suggest), or is it re-solved at each solver step with a short execution horizon as described in the RHC formulation?

2. Can you quantify how close the practical MPC-$\delta t$ policy is to exact value function-based optimal policy, at least on a small model where stronger baselines are feasible?

3. Better provide a table of runtime and peak VRAM on the same hardware for the large-model experiments, especially compared with FlowChef and any feasible OC baselines. The FLUX.2 results are promising, but explicit measurements would make the "practical and scalable on consumer hardware" claim much more convincing.

4. In Appendix B, the existence proof mixes the squared $L^2$ cost with the unsquared norm. in the final lower-semicontinuity step and the closing remark, the manuscript switches to $\int_0^1 \|u(t)\|dt$ instead of $\int_0^1\|u(t)\|^2dt$.

**Limitations:**

Yes

**Strengths And Weaknesses:**

Soundness. The formulation is generally sensible and well motivated. My main concern is that the theory is stronger than what is actually implemented. In practice, MPC-RHC is only approximately solved after coarse discretization, while MPC-Δt relies on a heuristic one-step approximation of the value function, so the theoretical guarantees do not directly apply to the practical algorithms. The CT comparison to “global control” should also be interpreted cautiously: the baseline itself is obtained via non-convex optimization and may only reach a local minimum. I also found a potentially important ambiguity around the $K=1$ variant. The main text describes RHC as replanning while executing only a short horizon, but Algorithm 3 in the appendix appears to set $\delta=1-t'$, effectively jumping directly to the terminal time. Since the $K=1$ case underlies the scalability claim for large models, this point should be clarified. Finally, there are a few minor technical and presentation inconsistencies in Appendix B that would benefit from cleanup.

Presentation. The paper is generally clear and well written.

Significance. The problem is important, and the paper’s strongest contribution is not "best restoration accuracy" but rather a more practical and scalable optimal-control-based guidance framework for flow models. The empirical results support “better than prior OC baselines and scalable to large models,” but provide weaker evidence for being the best practical inverse-problem solver overall.

Originality. The originality is moderate but well motivated. Using MPC as the organizing framework for conditional flow guidance is novel in this context and conceptually helpful, and the identification of a no-backprop $K=1$ regime is practically relevant. That said, the formal results largely follow standard dynamic-programming arguments, and the practical $K=1$ method appears conceptually close to FlowChef with explicit path regularization. In that sense, the contribution is less about new control theory and more about a useful reframing and a practical algorithmic spectrum exposing quality–memory–runtime trade-offs. This is still valuable, but the paper should position the novelty more explicitly in those terms.

Overall, I find the paper promising and potentially impactful, but my confidence would increase substantially if the authors close the theory-practice gap more explicitly and resolve the $K=1$ algorithmic ambiguity.

---

> ### Author Rebuttal · Authors · 2026-03-30
>
> Thank you for this careful review. We agree with the core points: the main contribution is the MPC-based framework and its practical quality–compute trade-offs, rather than new control-theoretic results. In the revision, we will (i) clarify the implemented MPC-RHC $(K=1)$ algorithm, (ii) make the approximation gap between theory and practice explicit, (iii) add quantitative approximation gap experiments, and (iv) report runtime/VRAM for the large-model setting.
>
> ---
>
> **Novelty Positioning**: We agree that the main novelty is MPC's use as a unifying framework for training-free control of flow models, together with the spectrum of algorithms exposing quality–compute trade-offs. We will revise the positioning to centre this contribution.
>
> **Theory–Practice Gap**: We agree that the theoretical guarantees apply to the continuous-time MPC formulation, while implemented methods introduce approximations. In the revision, we will make the main sources explicit:
> (i) time discretisation of the controlled dynamics via Euler integration (including piecewise-constant controls);
> (ii) inexact numerical solution of each MPC subproblem;
> (iii) for MPC-$\Delta t$, approximation of the value function with a one-step surrogate.
>
> We will clarify that Theorems 3.1–3.2 characterise limiting optimal policies rather than provide guarantees for the discretised implementations.
>
> In particular, for MPC-RHC the parameter $K$ controls the fidelity of this approximation: small $K$ (and especially $K=1$) corresponds to a coarse discretisation of the remaining horizon, limiting integration accuracy and the expressivity of the control. This behaviour is consistent with our controlled experiments, where increasing $K$ reduces the gap to the (estimated) global-control solution.
>
> Discretisation error is inherent to using standard flow-based models. We emphasise it is not a fundamental limitation of our framework (using *mean flow models* [1] instead of classical flow matching could eliminate this approximation error).
>
> **Global Control Comparison:** We agree that the "global control" baseline for CT experiments may be inexact (as optimization may find a local minimum). We will clarify our comparisons are relative to this standard baseline, not to a certified global optimum. We will also explicitly reference the controlled toy setting where we study convergence to the optimal trajectory as discretisation is refined.
>
> ---
> ### Key Questions:
>
> **Q1. Ambiguity in $K=1$.** The reviewer is correct that Algorithm 3 is misleading: line 3 sets $\delta \leftarrow 1-t'$, implying the procedure should jump directly to terminal time. This is a presentation error and will be corrected.
>
> In our implementation of MPC-RHC with $K=1$, we re-solve the optimisation at every timestep and execute only a fixed step $\Delta t = 1/N$ before replanning. Thus, the method is a receding-horizon procedure with single-step planning, not a one-shot optimisation followed by a terminal update.
>
> We agree that this approach shares similarities to FlowChef or FlowDPS, both of which involve optimizing the current iterate $x_t$ (or its projection to the terminal state $x_1$). However, neither method incorporates a regularization term in this update. We will discuss these algorithmic similarities in the revised manuscript.
>
> **Q2. Error of MPC-$\Delta$t.** We have added further analysis building on our 2D example in Appendix C. We numerically solve the HJB equation, providing the value function and optimal control. This enables a direct comparison between the true value function and the one-step approximation used in MPC-$\Delta t$ (Eqn. 12). As expected the discrepancy between the true value function and the approximation is largest early in the trajectory and decreases as $t \to 1$. We compare the trajectory generated by the MPC-$\Delta t$ policy with that obtained from the value function. The results are shown in this figure: https://bashify.io/i/vmvZwM
>
> **Q3. Runtime/Memory for FLUX.2:** We now provide an additional Table validating the memory and runtime of our method and comparators for the FLUX.2 experiments.
>
> | Method | Peak VRAM (GB) | Time (s) |
> |--|--|--|
> | Unconditional | 17.7 | 53 |
> | FlowChef | 18.4 | 92 |
> | MPC-RHC (K=1) | 18.4 | 101 |
>
>
> **Q4. Squared Cost.** Thank you for catching this typo. We will correct this so that the proof consistently uses the squared $L^2$ cost; the argument is unchanged since the squared norm is weakly lower semicontinuous.
>
> [1] Geng et al. "Mean Flows for One-step Generative Modeling" (https://arxiv.org/pdf/2505.13447)
>
> ---
> ## Summary
> We appreciate these comments, which improved both clarity and positioning. In revision, we (i) correct the $K=1$ algorithmic description, (ii) explicitly characterise the approximation gap, (iii) add controlled experiments quantifying this gap, and (iv) strengthen empirical support for scalability with runtime/VRAM measurements. We believe these changes align the presentation more closely with the contribution.

---

> > ### Author Rebuttal · Reviewer_AEUf · 2026-04-05
> >
> > Thanks for the rebuttal. It is helpful and improves clarity, but does not materially change my evaluation. The authors explicitly position the contribution as the MPC-based framework and its quality–compute trade-offs. The response also confirms that the theoretical guarantees apply to the continuous-time formulation, while the practical methods rely on discretization and approximations, so the theory–practice gap remains.
> >
> > The clarification that Algorithm 3 was misleading is appreciated, but since the  $K=1$ case underlies the scalability claim, this remains a non-trivial concern. The added runtime/VRAM numbers support feasibility on 24GB hardware, but they show comparable memory and slightly slower runtime than FlowChef, strengthening feasibility more than demonstrating a clear advantage.
> >
> > Overall, the rebuttal improves presentation and slightly increases confidence, but does not sufficiently address the main concerns to change my scores.

---

> > > ### Author Response · Authors · 2026-04-06
> > >
> > > Thank you for the follow-up and for acknowledging the improved clarity of our rebuttal. Please allow us to briefly summarise our position on these remaining points:
> > >
> > > ---
> > >
> > > *Theory-Practice Gap*: We agree that Theorems 3.1 and 3.2 apply to the continuous-time formulation. However, the discretisations used in our practical algorithms are not arbitrary: they are structured approximations of the underlying control problem that preserve the key design elements of the formulation. In particular, they retain the explicit control objective and regularisation, which in our experiments translate into improved stability and reconstruction fidelity.
> > >
> > > ---
> > >
> > > *Scalability and the FlowChef Comparison:* Our empirical results show MPC-RHC (K=1) scales efficiently to large-scale flow models. You rightly note that MPC-RHC (K=1) has a comparable memory footprint to FlowChef but is slightly slower. The increased runtime relative to FlowChef arises from the inclusion of the control regularisation term in the optimisation; this additional computation is directly tied to enforcing controlled deviations from the unconditional trajectory, and we observe that its inclusion yields substantially improved reconstruction fidelity in inverse problems. Our empirical results on FLUX.2 super-resolution (referenced in our response to Reviewer QPAM, https://bashify.io/i/6ry4f0) demonstrate this:
> > > - In a comparison of Pareto frontiers (PSNR vs. LPIPS), MPC-RHC consistently outperforms FlowChef across hyperparameter settings.
> > > - When choosing hyperparameters to maximize PSNR for both methods, MPC-RHC achieves ~30.8 dB compared to ~28.3 dB for FlowChef.
> > >
> > > ---
> > >
> > > Finally, we appreciate your assessment that using MPC as an organising framework is 'novel in this context and conceptually helpful.' By providing this unifying perspective, we show that existing guidance rules can be recovered as specific cases of our objective and we will discuss this more in the revised manuscript. This supports our framing of MPC as a broader control-based framework rather than a single algorithmic variant.

---

### Decision · Program_Chairs · 2026-04-30

**Decision:**

Accept (regular)

**Comment:**

This paper introduces MPC-Flow, a framework that applies Model Predictive Control (MPC) to flow-based generative models to solve inverse problems through a sequence of control sub-problems. Reviewers found the framework conceptually novel and appreciated its ability to scale training-free guidance to large-scale architectures like FLUX.2 (32B) on consumer hardware. During the rebuttal, the authors effectively addressed concerns regarding the theory-practice gap by making discretization approximations explicit and clarifying the implementation of the efficient $K=1$ variant. Furthermore, the authors strengthened the empirical evaluation by adding comparisons with recent solvers like FlowChef and FlowDPS, demonstrating a superior Pareto frontier between reconstruction fidelity and perceptual quality. While some minor data-labeling oversights were noted during the discussion, the reviewers agreed that the core technical contributions are robust and the accuracy–compute trade-off is well-validated. Given the positive ratings by majority of the reviewers and the practical use of the proposed scalable control method, the submission is recommended for acceptance.